# A geography of drought indices: mismatch between indicators of drought and its impacts on water- and food securities

Sarra Kchouk[1], Lieke A. Melsen[2], David W. Walker[1], Pieter R. van Oel[1]

[1]Water Resources Management Group, Wageningen University, Wageningen, 6708PB, The Netherlands
5  [2]Hydrology and Quantitative Water Management Group, Wageningen University, Wageningen, 6708PB, The Netherlands

*Correspondence to*: Sarra Kchouk (sarra.kchouk@wur.nl)

**Abstract.** Drought monitoring and Early Warning Systems (DEWS) are seen as helpful tools to tackle drought at an early stage and reduce the possibility of harm or loss. They usually include indices attributed to meteorological, agricultural and/or hydrological drought: physically based drought drivers. These indices are used to determine the onset, end and severity of a 10 drought event. Drought impacts, like water and food securities, are less monitored or even not included in DEWS. Therefore, the likelihood of experiencing these impacts is often simply linearly linked to drivers of drought. The aim of this study is to evaluate the validity of the assumed direct linkage between drivers of drought and water and food insecurities impacts of drought. We reviewed scientific literature on both drivers and impacts of drought. We conducted a bibliometric analysis based on 5000+ scientific studies in which selected drought indices (drivers) and drought related water and food insecurities (impacts) 15 were mentioned in relation to a geographic area. Our review shows that there is a tendency in scientific literature to focus on drivers of drought, with the preferred use of meteorological and remotely sensed drought indices. Studies reporting drought impacts are more localised, with relatively many studies focusing on Sub-Saharan Africa and Australasia for impacts with regard to food security and water security, respectively. Our review further suggests that studies of food and water insecurities impacts related to drought are dependent on both the physical and human processes occurring in the geographic area, i.e. the 20 local context. With the aim of increasing the relevance and utility of the information provided by DEWS, we argue in favour of additional consideration of drought impact indices oriented towards sustainable development and human welfare.

## 1 Introduction

Drought is a threat to a wide range of human activities in virtually all climate zones and countries (Van Loon et al., 2016a;Bachmair et al., 2016;Van Lanen et al., 2017). It is an elusive phenomenon without a clear onset and demise. In contrast 25 to other hazards such as floods, landslides or earthquakes, drought has a creeping nature causing impacts to persist for many years (Kim et al., 2019). Consequently, impacts can be cumulative for consecutive periods of droughts, devastating both ecosystems and societies (Bachmair et al., 2016;Van Lanen et al., 2017).

Many concepts exist for defining a drought (Santos Pereira et al., 2009;Lloyd-Hughes, 2014). Definitions of drought are either conceptual or operational. Conceptual definitions of drought are descriptive and highlight the natural hazard element: for

example, precipitation below what is expected or normal (Knutson et al., 1998). Operational definitions of drought highlight practical implications in an attempt to identify the onset, severity, and cessation of drought periods (Mishra and Singh, 2010). For example, the UN Convention to Combat Drought and Desertification (UN Secretariat General, 1994) defines drought as "*when precipitation has been significantly below normal recorded levels, causing serious hydrological imbalances that adversely affect land resource production systems*".

The numerical value of hydro-climatic variables is associated to three main types of drought: meteorological, agricultural (or soil moisture) and hydrological droughts. These variables are in fact drivers, which refer to the contributing or counteracting factors that affect the development of droughts (Seneviratne, 2012). Those drivers are used by many drought studies as the framework to represent drought propagation. In the literature, the temporal propagation of drought is often considered to be a sequence occurring in an almost linear order (Wilhite and Glantz, 1985;Zargar et al., 2011;Bachmair et al., 2016), and in which

humans have no direct influence. This is a simplification of a complex process, where it is considered that an anomaly (e.g. lower precipitation, higher temperature than average) of the values of those drivers will lead to a cascade reaction influencing the magnitude of other physical variables and leading in turn to the subsequent type of drought. As such, hydrological drought is inaccurately simplified as a result from the persistence in duration of agricultural (soil moisture) drought, which itself is simplistically attributed to the persistence of meteorological drought.

Drought monitoring and Early Warning Systems (DEWS) aim to monitor the drivers of drought to predict drought. They aim to tackle drought at an early stage to reduce the possibility of harm or loss. For assessing the severity of a drought, physical variables are usually translated into indices of drought. The difference between their values and the threshold used to define the level of dryness is considered to depict the severity of a drought (Vogt et al., 2018). Drought impacts, such as water- and food security, are rarely continuously monitored or even included in DEWS. This is understandable as there is already a

plethora of definitions for drought and drought types, and there are at least as many possibilities for defining impacts (Mishra and Singh, 2010;Wilhite, 2000;Santos Pereira et al., 2009). Drought impacts are non-structural, difficult to quantify or monetise, and can be direct or indirect due to the extended nature, in time and area, of drought (Wilhite et al., 2007;Logar and Van den Bergh, 2011;Bachmair et al., 2016). In addition, most the of DEWS do not take the underlying vulnerabilities of the drought affected or monitored areas into account. Thus, in the current configuration of most DEWS, the presumed likelihood

of experiencing impacts is mainly linked to the severity of climatic features only (e.g. :(Princeton Flood and Drought Monitors;U.S. Drought Monitor;Brazilian Drought Monitor)).

    This study aims to review scientific reporting on drought drivers and drought impacts for affected countries and analyse how these two compare. Improving our understanding of the linkage and separation between drought drivers and drought impacts enables us to provide directions to further improve the accuracy of the information provided by DEWS. We retrieved scientific

studies from countries in which selected drivers of drought and food and water securities impacts of drought are mentioned. The components of drought drivers and impacts on which the literature focused were explored and compared for different areas of the world.

**2 Data and Methods**

**2.1 Methodological approach**

The methodological approach comprises three steps:

Step 1. Exploring which drought drivers are the most recurrent in the scientific literature. We investigated which indices of drought drivers are most frequently used in scientific drought-related studies and to what drought type they were linked. For each of these scientific studies we also retrieved the country of focus. This allowed us to identify: the most frequently mentioned type of drought for different geographic regions, and the prevalent drought indices used in scientific studies.

Step 2. Exploring which drought impacts are the most recurrent in the scientific literature. In contrast with drought drivers, for drought impacts there are no established indices commonly used in DEWS and in scientific studies. We thus retrieved from scientific articles, keywords associated to drought impacts related to water security and food security. This allowed the identification of the most frequently mentioned water- and food-related drought impacts.

Step 3. Comparing the findings of Steps 1 and 2. This enabled evaluation of the alignment between reported drought types and
impacts, with regard to the number of publications and differences in geographic focus.

**2.2 Data**

We considered the number of studies about drought indices and drought impacts, respectively, and their geographical distribution as our units. Our list of drought indices is based on two prominent studies in the field of drought indices: indices commonly used operationally to depict different types of drought (Svoboda and Fuchs, 2016) and the indices commonly used
by water managers (Bachmair et al. (2016). Our list will, however, inherently be incomplete because many other indicators exist beyond the ones mentioned in these two studies. This resulted in 32 indices that we linked to three main drought types (Table 1): meteorological (9 indices), soil moisture/agricultural (15) and hydrological (8) drought.

| Meteorological Drought indices studies | Total number of studies of drought indices : 5567 | | Total number of studies mentioning a country: 4023 | | Studies not mentioning a country : 27.7% | Top 3 subject area retrieved from Scopus |
|---|---|---|---|---|---|---|
| "Meteorological drought" Indices mentioned in the study | Acronym | Input data | Number of studies | Studies mentioning a country | Portion of studies not mentioning a country (%) | |
| Standardized Precipitation Index | SPI | Precipitation | 2451 | 1812 | 26.1 | 1) Environmental Science 2) Earth and Planetary Sciences 3) Agricultural and Biological Sciences |

| Index | Abbr | Input data | Number of studies | Studies mentioning a country | Portion of studies not mentioning a country (%) | Top 3 subject area |
|---|---|---|---|---|---|---|
| Standardized Precipitation Evapotranspiration Index | SPEI | Precipitation, temperature | 1059 | 751 | 29 | 1) Environmental Science<br>2) Earth and Planetary Sciences<br>3) Agricultural and Biological Sciences |
| Aridity Index | AI | Precipitation, temperature | 247 | 182 | 26.3 | 1) Environmental Science<br>2) Earth and Planetary Sciences<br>3) Agricultural and Biological Sciences |
| Precipitation Deciles | Deciles | Precipitation | 12 | 9 | 25 | 1) Earth and Planetary Sciences<br>2) Environmental Science<br>3) Engineering |
| Keetch-Byram Drought Index | KBDI | Precipitation, temperature | 84 | 66 | 21.4 | 1) Environmental Science<br>2) Agricultural and Biological Sciences<br>3) Earth and Planetary Sciences |
| Palmer Drought Severity Index | PDSI | precipitation, temperature, available water content | 1279 | 867 | 32.2 | 1) Environmental Science<br>2) Earth and Planetary Sciences<br>3) Agricultural and Biological Sciences |
| Percent of Normal Precipitation (Index) | PNPI | Precipitation | 23 | 18 | 21.7 | 1) Environmental Science<br>2) Earth and Planetary Sciences<br>3) Agricultural and Biological Sciences |
| Rainfall Anomaly Index | RAI | Precipitation | 304 | 244 | 19.7 | 1) Earth and Planetary Sciences<br>2) Environmental Science<br>3) Agricultural and Biological Sciences |
| Self-Calibrated Palmer Drought Severity Index | scPDSI | Precipitation, temperature, available water content | 108 | 74 | 31.5 | 1) Earth and Planetary Sciences<br>2) Environmental Science<br>3) Agricultural and Biological Sciences |
| **Agricultural and Soil Moisture Drought indices studies** | **Total number of studies of drought indices : 5085** | | **Total number of studies mentioning a country: 3137** | | **Studies not mentioning a country : 38.3%** | **Top 3 subject area** |
| **"Agricultural drought" Indices mentioned in the study** | **Input data** | | **Number of studies** | **Studies mentioning a country** | **Portion of studies not mentioning a country (%)** | |

| | | | | | | |
|---|---|---|---|---|---|---|
| Crop Moisture Index | CMI | precipitation, temperature | 43 | 20 | 53.5 | 1) Earth and Planetary Sciences<br>2) Agricultural and Biological Sciences<br>3) Environmental Science |
| Evaporative Stress Index | ESI | Remotely sensed potential evapotranspiration | 88 | 42 | 53.3 | 1) Agricultural and Biological Sciences<br>2) Earth and Planetary Sciences<br>3) Environmental Science |
| Evapotranspiration Deficit Index | ETDI | soil water in the root zone on a weekly basis, which is computed from SWAT model | 17 | 13 | 23.5 | 1) Environmental Science<br>2) Earth and Planetary Sciences<br>3) Agricultural and Biological Sciences |
| Enhanced Vegetation Index | EVI | NIR/red/blue surface reflectances, canopy background adjustment, coefficients of the aerosol resistance for correction for aerosol influences in the red band. | 305 | 206 | 32.2 | 1) Earth and Planetary Sciences<br>2) Environmental Science<br>3) Agricultural and Biological Sciences |
| Normalized Difference Vegetation Index | NDVI | Spectral reflectance measurements acquired in the red and near-infrared regions | 2041 | 1288 | 36.9 | 1) Earth and Planetary Sciences<br>2) Environmental Science<br>3) Agricultural and Biological Sciences |
| Leaf Area Index | LAI | Leaf and ground area | 1152 | 583 | 49.4 | 1) Agricultural and Biological Sciences<br>2) Environmental Science<br>3) Earth and Planetary Sciences |
| Palmer Moisture Anomaly Index – known as the Palmer Z index | PZI | Derivative of the PDSI calculation precipitation, temperature, available water content | 47 | 30 | 36.2 | 1) Earth and Planetary Sciences<br>2) Environmental Science<br>3) Agricultural and Biological Sciences |
| Soil Adjusted Vegetation Index | SAVI | Spectral reflectance measurements acquired in the red and near-infrared regions, with the addition of a soil brightness correction factor | 68 | 37 | 45.6 | 1) Agricultural and Biological Sciences<br>2) Environmental Science<br>3) Earth and Planetary Sciences |
| Soil Moisture Anomaly | SMA | precipitation, temperature, available water content | 138 | 87 | 37.0 | 1) Earth and Planetary Sciences<br>2) Environmental Science<br>3) Agricultural and Biological Sciences |

| Soil Moisture Deficit Index | SMDI | soil water in the root zone on a weekly basis, which is computed from SWAT model | 13 | 10 | 23.1 | 1) Environmental Science<br>2) Earth and Planetary Sciences<br>3) Agricultural and Biological Sciences |
|---|---|---|---|---|---|---|
| Soil Water Deficit Index | SWDI | | 33 | 26 | 21.2 | 1) Earth and Planetary Sciences<br>2) Environmental Science<br>3) Agricultural and Biological Sciences |
| Soil Water Storage | SWS | available water content, reservoir, soil type, soil water deficit | 717 | 494 | 31.1 | 1) Agricultural and Biological Sciences<br>2) Environmental Science<br>3) Earth and Planetary Sciences |
| Vegetation Condition Index | VCI | (same as) NDVI | 271 | 187 | 30.1 | 1) Earth and Planetary Sciences<br>2) Environmental Science<br>3) Computer Science |
| Vegetation Drought Response Index | VegDRI | SPI, PDSI, percentage annual seasonal greenness, start of season anomaly,<br>land cover, soil available water capacity, irrigated agriculture and defined ecological regions | 14 | 13 | 7.1 | 1) Earth and Planetary Sciences<br>2) Environmental Science<br>3) Agricultural and Biological Sciences |
| Vegetation Health Index | VHI | NDVI and brightness temperature, both from thermal bands | 138 | 101 | 26.8 | 1) Earth and Planetary Sciences<br>2) Environmental Science<br>3) Computer Science |
| **Hydrological Drought indices studies** | **Total number of studies of drought indices : 550** | | **Total number of studies mentioning a country: 344** | | **Studies not mentioning a country : 37.5%** | **Top 3 subject area** |
| **"Hydrological drought" Indices mentioned in the study** | | **Input data** | **Number of studies** | **Studies mentioning a country** | **Portion of studies not mentioning a country (%)** | |
| Reservoir Level | | Water levels in reservoirs | 72 | 35 | 51.4 | 1) Environmental Science<br>2) Engineering<br>3) Earth and Planetary Sciences |
| Palmer Hydrological Drought Index (PHDI) | PHDI | precipitation, temperature, available water content | 58 | 34 | 41.4 | 1) Environmental Science<br>2) Earth and Planetary Sciences<br>3) Agricultural and Biological Sciences |

| | | | Number of studies | Studies mentioning a country | Portion of studies not mentioning a country (%) | |
|---|---|---|---|---|---|---|
| Streamflow Drought Index | SDI | Streamflow values | 180 | 117 | 35 | 1) Environmental Science<br>2) Agricultural and Biological Sciences<br>3) Earth and Planetary Sciences |
| Standardized Runoff Index | SRI | *"Runoff"* | 106 | 69 | 34.9 | 1) Environmental Science<br>2) Earth and Planetary Sciences<br>3) Engineering |
| Standardized Streamflow Index | SSFI | Streamflow data | 85 | 56 | 34.1 | 1) Environmental Science<br>2) Earth and Planetary Sciences<br>3) Agricultural and Biological Sciences |
| Streamflow anomaly | | Streamflow data | 9 | 8 | 11.1 | 1) Environmental Science<br>2) Earth and Planetary Sciences<br>3) Agricultural and Biological Sciences |
| Standardized Water-level Index | SWI | Groundwater well levels | 17 | 13 | 23.5 | 1) Environmental Science<br>2) Earth and Planetary Sciences<br>3) Social Sciences |
| Surface Water Supply Index | SWSI | Reservoir storage, streamflow, snowpack and precipitation | 23 | 12 | 47.8 | 1) Environmental Science<br>2) Engineering<br>3) Social Sciences |
| **Drought impacts studies** | **Input data** | | **Number of studies** | **Studies mentioning a country** | **Portion of studies not mentioning a country (%)** | |
| **Food security** | Food security, famine, hunger, malnourishment, malnutrition, agricultural loss. | | 4764 | 2601 | 45.4 | 1) Agricultural and Biological Sciences<br>2) Environmental Science<br>3) Social Sciences |
| **Water security** | Water security, water access, water availability, water crisis | | 805 | 506 | 37.1 | 1) Environmental Science<br>2) Social Sciences<br>3) Earth and Planetary Sciences |

**Table 1: Table of the drought indices and impacts sought in studies retrieved from Scopus. Their acronym, input data when applicable, total number of studies and number of studies mentioning a country, are detailed.**

We opted for Scopus to retrieve the scientific publications of interest as it is the database covering the largest range of both, peer-reviewed literature type (scientific journals, books and conference proceedings), and disciplinary fields (science,

technology, medicine, social sciences, and arts and humanities ) (Scopus, 2021). We then searched in the Scopus database for queries strictly including "drought" AND "[the indicator]" in the title, abstract and authors' keywords of the studies. We repeated the queries for each indicator individually as we were interested in knowing country-based preferences. The sum of the individual indices linked to drought queries returned 4137 articles for the "meteorological" drought type of indices, 2799 articles linked to "agricultural" drought and 393 articles linked to "hydrological" drought. The title, authors, author's keywords, year of publication, journal name and abstract were retrieved using the Bibliometrix package (Aria and Cuccurullo, 2017) executed on R (version 4.0.0) following Addor and Melsen (2019). In the title, keywords and abstract of each paper, names of countries were identified, corresponding to the area of application of the study. The same approach was followed for the drought impacts. We grouped drought impacts into two focus categories: food security and water security. Their keywords are indicated in Table 1. The queries included "drought" AND selected "[drought impact]". This resulted in 4764 articles linking drought to food security and 805 articles linking drought to water security.

All articles were published between 1960 and March 2021 and the exact queries for both drought indices and impacts are included in Table A1. Even though we recognise drought can impact ecosystems, this topic was excluded from the analysis for reasons of brevity. The dataset and the script used for its analysis are both available for consultation (Kchouk et al., 2021).

Many scientific studies are methodological; their goal can be the validation, calibration or improvement of the indices, thus, not all studies have a focus country. We only considered studies mentioning a country in their title, abstract and keywords; this being the only criteria of inclusion or rejection of papers in our analysis. This reduced the number of studies including a name of a country in their title, abstract and keywords by 28% for drought indices and by 44% for drought impacts. We also did a manual verification on some of the scientific studies to see if the association with a country was valid. This allowed us to bring some corrections to the metadata to avoid incorrect associations (e.g. removing mentions of the "Indian Ocean" that led to the incorrect association of the studies to India ; removing the copyrights, generally in the end of the abstract, referring to another country than the one of the study).

## 3 Results

### 3.1 Drought types and indices

The indices mentioned in the drought-related studies were classified according to the categories used in Table 1; their frequency of occurrence is shown in Fig. 1. Meteorological drought indices are reported most frequently, followed by agricultural or soil moisture drought indices, and hydrological drought indices. The most frequently mentioned indicator is the Standardised Precipitation Index (SPI), followed by the Normalised Difference Vegetation Index (NDVI). Hydrological drought indices are less frequently utilised in comparison to the two other categories.

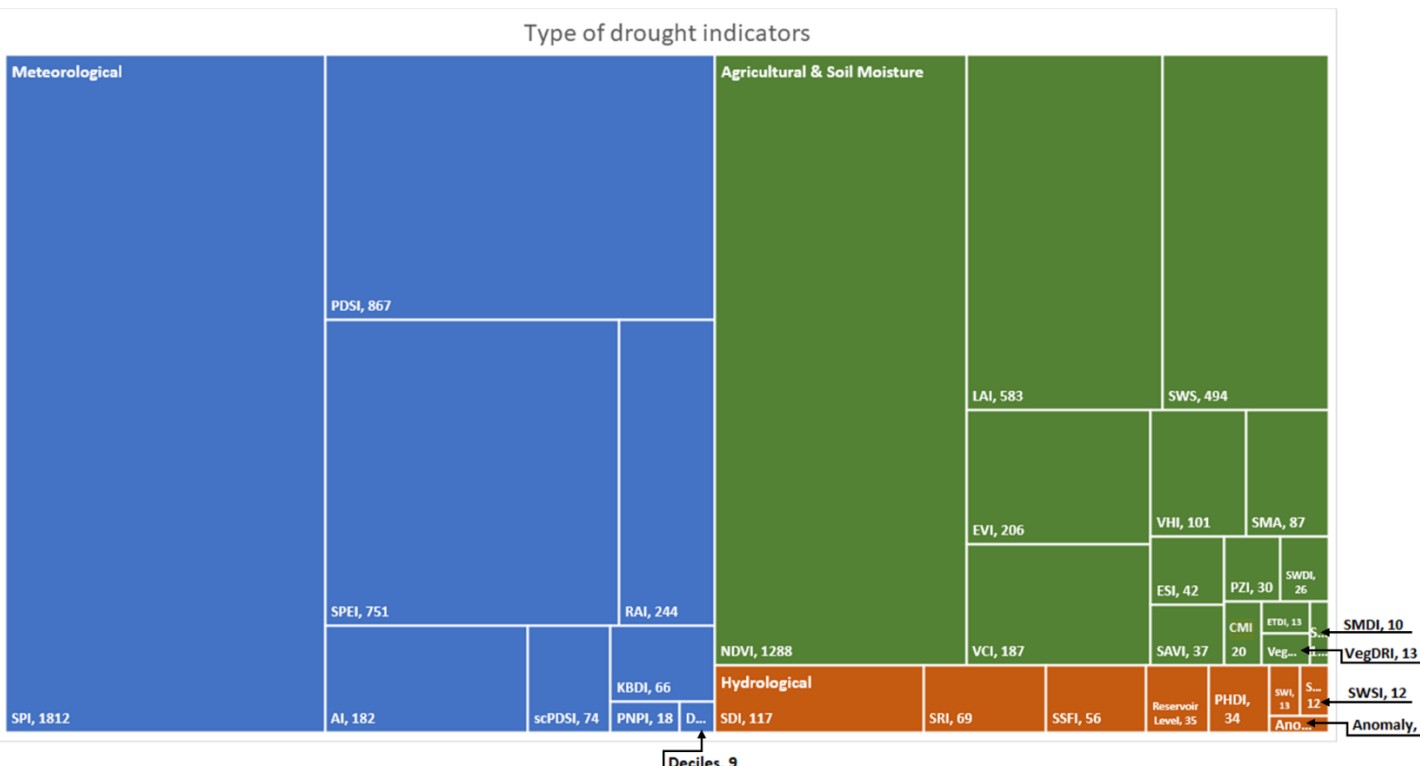

**Figure 01: Treemap showing the proportion of indices for different drought types (blue is meteorological, green is agricultural and soil moisture drought, and orange is hydrological drought) employed in the title, abstract and keywords of drought-related studies on Scopus. The number indicates the number of studies including a country in their title, abstract or authors' keywords.**

For the regions of Australia-Oceania, Middle-East and North Africa (MENA) and Sub-Saharan Africa (SSA), there are fewer studies utilising hydrological drought indices than for the other regions (Fig. 2). Further geographical differences are observed from Fig. 2. Most areas resemble the overall pattern shown in Fig. 1; exceptions are Australia-Oceania and Sub-Saharan Africa, where agricultural drought indices are most frequently reported.

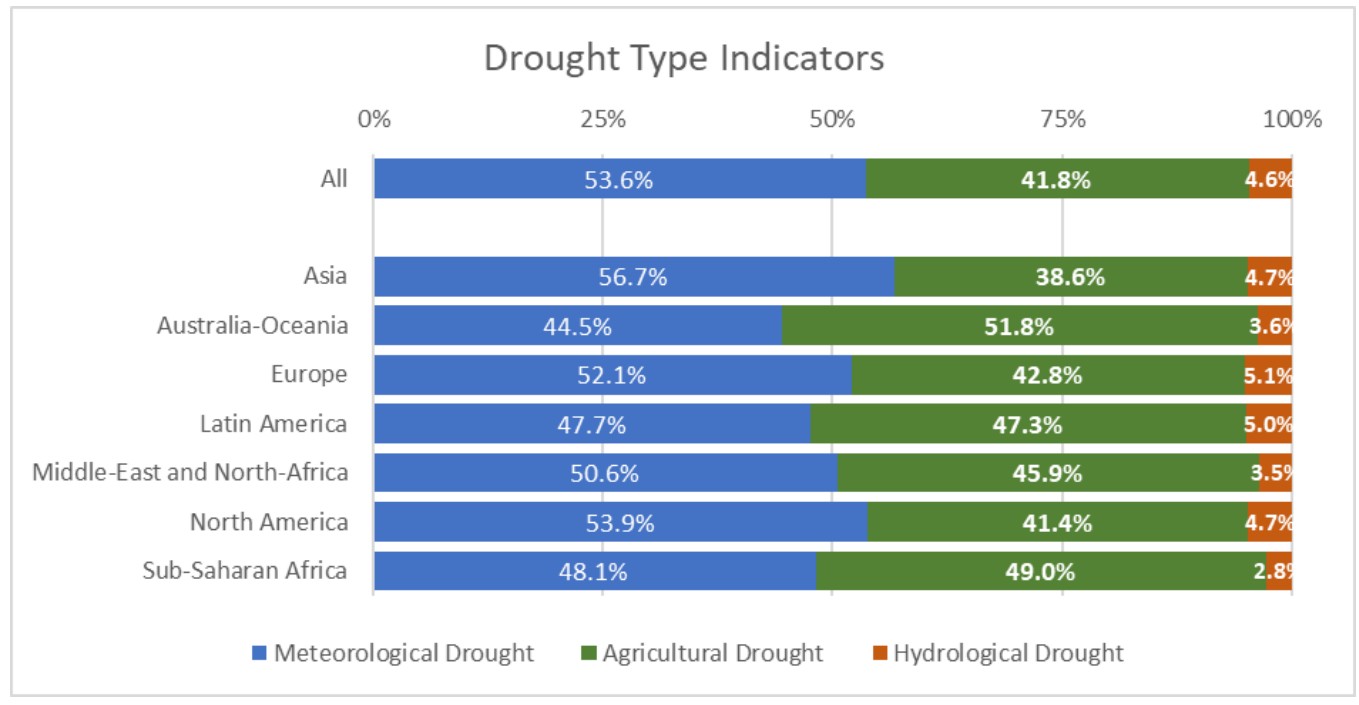


**Figure 02: Barplot showing the proportion of drought type studies per region of the world, according to the drought indices referred to in the title, abstract and keywords of drought-related studies on Scopus.**

In addition, not only are meteorological drought indices the most investigated, they are also the most associated with a country in studies, in comparison to agricultural drought, hydrological drought and impacts (Table 1). Meteorological drought indices

represents 53 % of the scientific studies while agricultural drought represents 42 % and hydrological drought , only 5 %. This indicates that in most of the studies, rainfall and the temperature are the dominant criteria utilised to report the occurrence of drought. Such a result is expected because of the ease of use of meteorological drought indices. We further develop this point in Sect. 4.3.

During the preliminary research that lead to the results mentioned in our study, we conducted a time analysis. We visualised

and compared the evolution of the usage of drought indices and drought impacts in the literature in order to analyse and link it to factors such as improved data availability, scientific progress or a change in the societal view on droughts (not shown). However, we did not find any remarkable pattern, peak or correlation. Therefore, we decided to not include this part in our study.

### 3.2 Drought-related impacts: food security and water security

Globally, there were five times more studies linking drought to food-security than drought to water-security (Fig. 3). This pattern is the same for most areas of the world. For Sub-Saharan Africa the predominance of food security indices is most pronounced (93%), followed by Asia and Europe (84%). Australia-Oceania is the only region where drought-related water

security studies predominate over food security studies (52%), while Sub-Saharan Africa is the region where it is reported the least (6.6%).

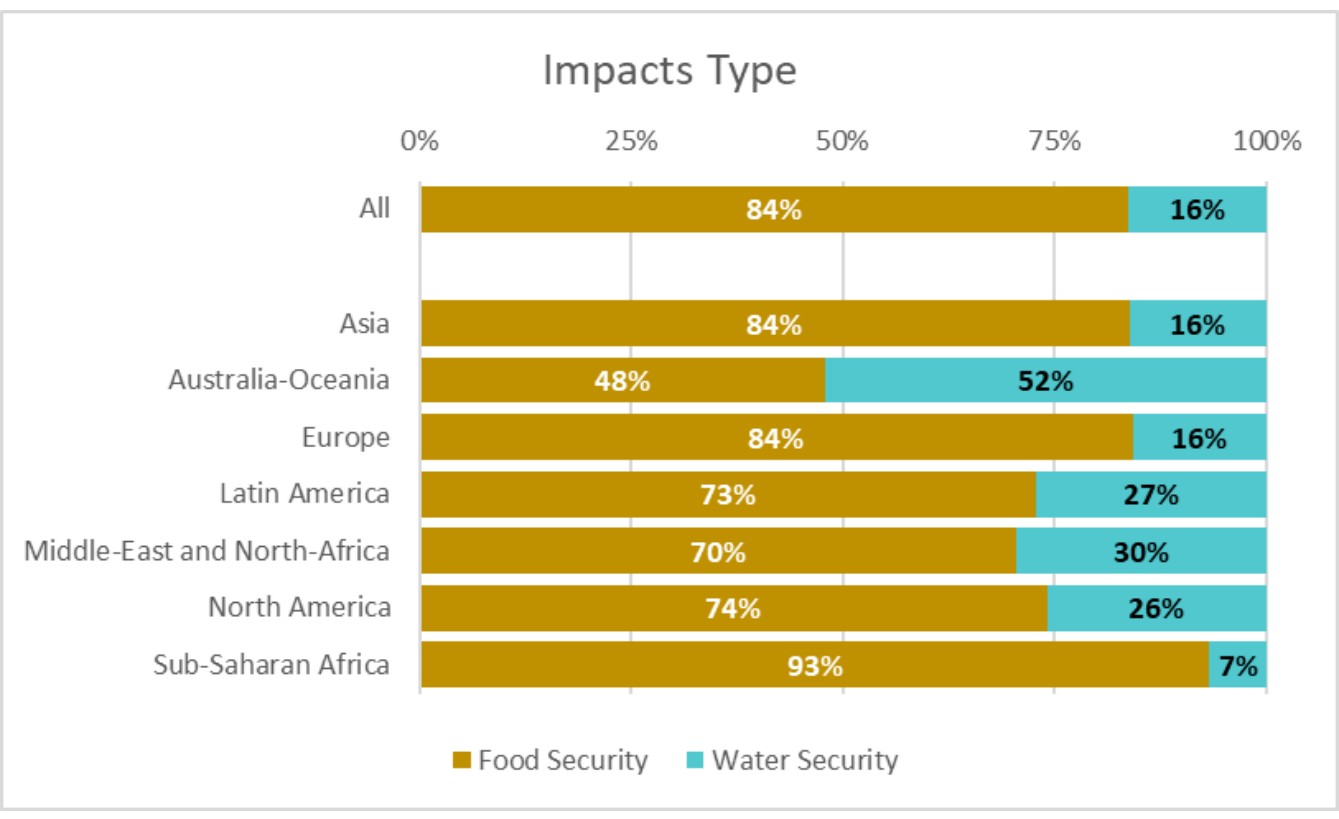


**Figure 03: Barplot showing the proportion of food and water security studies related to drought per region of the world on Scopus.**

### 3.3 Geographic patterns for indices of drivers and impacts

Figure 4 shows that drought-drivers studies are quite evenly distributed across the regions except for SSA. The height of the dark blue boxes is substantially smaller than the others, suggesting that the share of SSA in drought-drivers studies is minor.

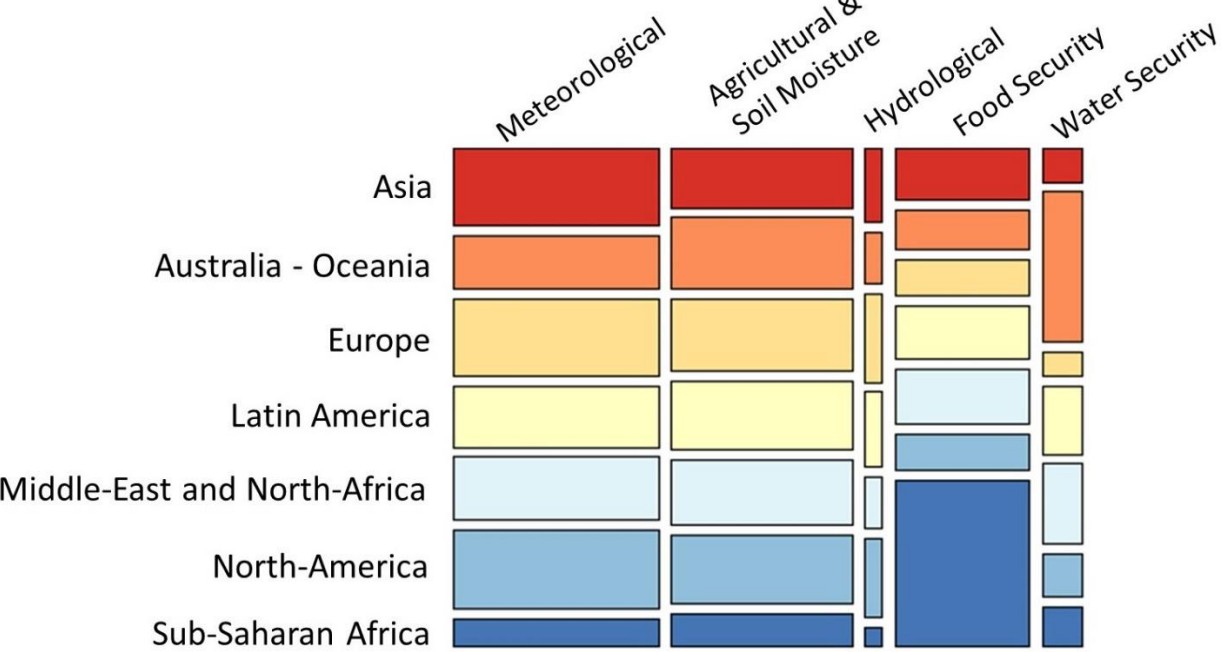


**Figure 04: Mosaic plot showing how frequently keywords, being the types of drought and impacts, were mentioned in the titles, abstracts, and keywords in drought-related studies on Scopus. The height (vertical) of each box indicates how frequently the keyword is used for each region (the frequency was scaled by the number of papers for each region, that is, the plots show the keyword frequency if all the regions had an equal number of papers). The width (horizontal) of each box indicates the relative frequency of**

**each keyword.**

In the same way, two geographical patterns appear in the share of drought-related impacts studies. The height of the boxes of SSA and Australia-Oceania for food and water securities, respectively, related to drought is significantly larger than those of the other regions for the same indicator category. This means that food security related to drought is most frequently reported for SSA and that water security related to drought is most frequently reported for Australia-Oceania. Similarly, drought-related

water security is least reported for Europe.

The geographical pattern of drought drivers and impact studies seen in Fig.4 is also present in the cartogram representations in Fig. 5. In this cartogram representation, each country has been rescaled in proportion to the number of studies on Scopus related to drought indices or water and food security impacts. First, the three drought drivers categories appear to have the same pattern of investigation, all mostly focused on northern high-income countries. The United States and Mexico, North-

Mediterranean countries and Australia-Oceania are strongly focusing on drivers in drought-related studies. Middle-income countries with high demographic and economic growth such as China, India and Iran also see a focus on drought-related drivers. They stand out from their geographic neighbours that are almost disappearing from the map.

DRIVERS                                    IMPACTS

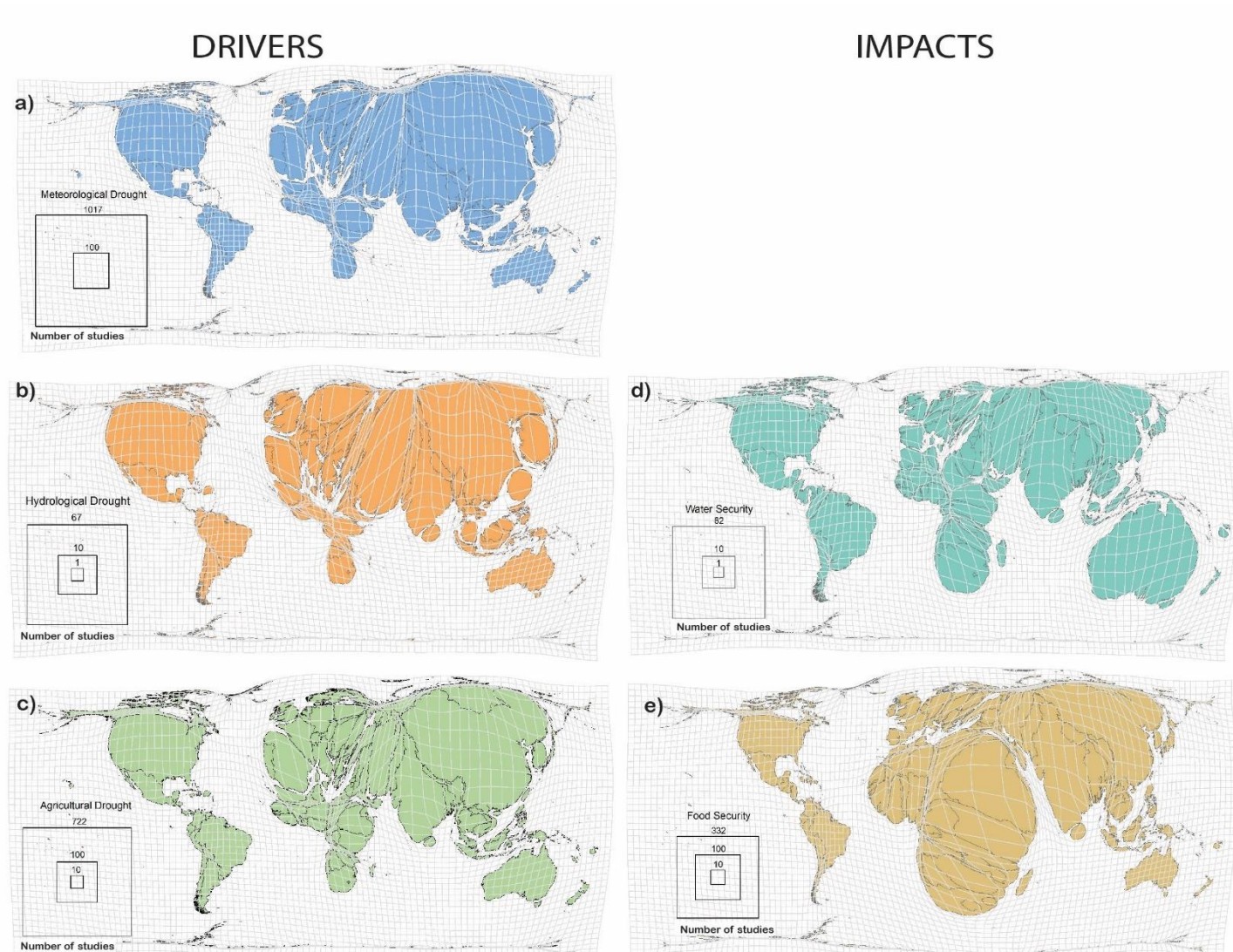


**Figure 05 : Contiguous cartograms (Gastner-Newman) of the world with each country rescaled in proportion to the number of studies on Scopus related to drought and a) Meteorological drought indices b) Hydrological drought indices c) Agricultural and Soil Moisture drought indices d) Food security e) Water security. The size of the square relates to the size of the countries and indicates the number of studies.**

In contrast, the African continent is strongly under-represented in terms of drought drivers studies, particularly with regard to meteorological and hydrological drought indices, with notable exceptions for Ethiopia, Kenya and South Africa. However, the distribution of agricultural and soil moisture drought studies appears to be more even in African countries, and higher in Sahelian countries.

Looking at the geographical repartition of drought-related impacts studies (Fig. 5d and 5e), two main observations are notable. First, the repartition of the impacts studies differs from the drivers studies. Second, both impacts, food and water security, show a different geographic pattern. Water security related to drought is most frequently investigated for Australia, the USA and Mexico, Brazil, the Middle East and South Africa. In contrast, food security is most commonly investigated for India, Ethiopia, Kenya and other African countries.

## 4 Discussion

This bibliometric study shows that unbalanced attention is given to drought drivers and impacts across the world. In this discussion section, we start by raising four hypotheses to explain why some features of drought are more frequently reported for some regions or countries than for others. The four hypotheses relate to: physical conditions (Sect. 4.1), socio-economic conditions (Sect. 4.2), data availability (Sect. 4.3), and scientific interests and orientation (Sect. 4.4). We continue by discussing potential limitations in our methodological approach (Sect. 4.5). We posit that these four hypotheses are also the four dimensions that are inherent to the local context of a geographic area. Drought monitoring is influenced by these to accurately predict droughts, their severity and impacts. In that sense, we end by formulating recommendations (Sect. 4.6) about shifting the scope of drought metrics to match the local context of a specific drought event.

### 4.1 Physical conditions

The most notable result from Sect. 3 is the more abundant investigation of meteorological drought over agricultural drought and hydrological drought (except in SSA and Australia-Oceania), with the SPI being the most used indicator in drought-related studies.

By focusing on meteorological drought, it is mainly the deficit of precipitation that is investigated. In humid areas, tropical, continental or temperate climates, a deficit of precipitation is less likely to affect the overall physical water scarcity and cause water shortage. In that sense, the occurrence of a drought is only statistically-based and not reflecting a true water deficit for the demand, only a below average situation (which is, however, in line with formal definitions of drought). In arid and semi-arid climates with lower levels of precipitation, it is recommended to use SPI cautiously because it can fail to indicate drought occurrence (Wu et al., 2007) and opt instead for indices that include evapotranspiration like the SPEI (Salimi et al., 2021). In such areas where evapotranspiration plays a larger role with regard to evaporative demand, water shortage is more common. For arid and semi-arid areas with low average rainfall and a higher risk of water scarcity, it may be more appropriate to determine water deficit at the crop, field or farm scale. This could explain the more frequent use of agricultural drought indices in the more arid Australian-Oceanian and Sub-Saharan regions (Fig. 2 & 4) that mainly monitor vegetation (NDVI, LAI) and soil water content (SWS) (Fig.1).

For some agricultural drought indices, there is both an upper and a lower limit that is independent of whether the climate of the area is arid or humid: vegetation health or soil water content are or are not frequently deteriorated or in deficit, respectively.

In that sense, agricultural drought indices are relevant for any type of climate. However, SPI and most MD and hydrological drought indices, are statistical values showing a deviation from average and standardised for all climates. Even if they remain meaningful, drought is more challenging in dry climates rather than wet climates. This key point is dismissed because of the statistical and standardising propensity of meteorological drought and hydrological drought indices, in contrast to the values of agricultural drought indices that are a practical interpretation of hydro-climatic features (e.g. of the reflectance, in the case

of NDVI and LAI).

**4.2 Socio-economic conditions**

SSA combines the lowest number of studies about drought indices with the highest proportion in terms of drought impacts (Fig. 4). Even though SSA is known to experience a rise of temperatures and an increase of aridity in the past, present and future by observation and model projections (Niang et al., 2014;Serdeczny et al., 2017) the reported impacts in the Emergency

Database (EM-DAT) are scarce (Harrington and Otto, 2020). Yet, the International Disaster Database (EM-DAT) run by the Centre for Research on the Epidemiology of Disasters (CRED), has the most complete and global records of past natural and human-made disasters events (Guha-Sapir et al., 2012).

Most of SSA is in a situation of economic water scarcity (Molden, 2013), implying a lack of human, institutional and financial capital to satisfy the demand for water, even in areas where the physical availability of water is not limited. The symptoms

described by Molden (2013) associated to economic water scarcity include scant infrastructure development, either small or large scale, meaning that populations experience difficulties obtaining sufficient water to meet agricultural or domestic needs. Applying the same reasoning, drought mitigation or monitoring bodies and scientific publications are a product of human, institutional and financial capital. Thus, it is likely that drought drivers are under-investigated in SSA, leading to the same effects of economic water scarcity: water and food insecurities. Also, the report of impacts of extreme weather in SSA to

disaster databases as EM-DATA are predominantly conducted by non-governmental organisations rather than governments, often as a side product of their main task to identify the location with the greatest need for humanitarian aid (Harrington and Otto, 2020).

In some areas, food insecurity can be a cumulative result of a dry climate and high pressure on natural resources enhanced by rapid demographic growth. Countries such as Bangladesh, China, Ethiopia, India, Indonesia and Pakistan, have some of the

highest number of drought-related food security publications (Fig. 5). Most of these countries have high fertility rates and rapid population growth (United Nations, 2019;Vollset et al., 2020). According to the Food and Agriculture Organization (FAO (2010)), the majority of the world's undernourished people live in these six countries and over 40% live in China and India alone. The same applies for the countries of SSA, presenting the highest population growth rate in the world (World Bank, 2019), the highest number of drought-related food security publications (Fig. 5), and 22% of the population being

undernourished (FAO et al., 2019). A rapid population growth increases the challenge of adequately meeting nutritional needs

as food production depends on croplands and water supply, which are under strain as human populations increase. This suggests that countries with arid climates and a high population growth are more exposed to food security impacts.

Moreover, populations of low income countries are the most exposed to drought-related food insecurity. In the world's poorest countries, around 30 percent of GDP comes from agriculture; those countries are mostly concentrated around the Sahelian region: Mali (37.4% of GDP), Niger (35.4%), Chad (46.1%), Central African Republic (31.9%), Sudan (31.2%), Kenya (31.1%) and Ethiopia (34.7%) (World Bank, 2016). As we can see from Fig.5, those countries are most commonly reporting food security impacts related to drought. In contrast, in OECD economies - regarded as developed and high-income countries – agriculture accounts for less than 1.5 percent of GDP (World Bank, 2016). In the same way, we note the fewest amount of publications related to food security in those OECD countries. Also, in these Sahelian countries, agriculture accounts for more than 80% of the livelihoods (FAO, 2021). As more people rely on agriculture for their livelihood, they are more exposed to hazards like drought and thus vulnerable to food-insecurity and the poverty trap.

It is also important to mention the link between food security and governance. Food security is dependent on a complex interplay of factors. Some are outside the direct control of governments, like hydrometeorological extremes. But institutions, rules and political processes do play an important role in reaching increased food security. According to the Food and Agriculture Organization (FAO, 2011), "*food security is unlikely to develop where there is not an organized, politically active and mobilized constituency pushing the issue higher on the public and political agenda*". Thus, good governance is crucial for reaching food security. Corruption is one of the pervasive aspects of bad governance. It can affect food security by creating inefficiencies in the use of natural resources and food distribution (Economist Intelligence Unit, 2015). Practices of corruption are spread in low-, middle- and high-income countries to different degrees (Transparency International, 2021) and in different levels of the food production and distribution chain (Transparency Int'l, 2019). Low-income countries are indeed the ones struggling the most to tackle corruption (Transparency International, 2021) contributing to their already prominent exposure to food insecurity. The addition of corruption, an indication of misallocation of resources and incapacity to successfully implement change and development, increases the risk of stagnation of food availability and indicates those countries as less suitable prospects for successful intervention (Economist Intelligence Unit, 2015).

In other words, focusing on physical drivers of drought is an advantage more apt to be of interest in areas where more basic and essential needs, such as food security, have been met.

### 4.3 Data availability

The SPI is the most widely used index in drought-related studies (Table 1 and Fig. 1). This can be explained by its ease of use: First, it only requires (monthly) precipitation data, easy to monitor by use of rainfall gauge networks or satellite estimation. Second, SPI reference values exist so they can be compared and are applicable in all climate regimes. Finally, SPI can be computed for different periods of time including periods of record containing missing-data, even though it ideally needs at least 30 years of monthly precipitation data (WMO, 2012).

However, all these strengths are at the same time weaknesses. The SPI will provide in all cases an output whatever inputs are used (Svoboda and Fuchs, 2016). As an example, a significant quantity of zero precipitation values at short time scales may

lead to biased values of the SPI, because the rainfall might not fit for the recommended gamma distribution, which is a fundamental first step of the SPI calculation (Wu et al., 2007). This scenario is applicable to dry climates with a distinct dry season when calculated for periods shorter than 12 months. As mentioned in section 4.1, an index including an additional temperature parameter to account for evapotranspiration is more suitable for such areas. As we can see in Figure 6, many countries with dry climates (Iran, Australia and Pakistan) commonly use the SPI in their drought-related studies. In those dry

contexts, it has been proposed to focus on the duration of the drought rather than only its severity (Wu et al., 2007). However, even short-lived dry spells often combined with heatwaves of a few days, characteristic of dry climates, when occurring during the reproductive stage of crop development can be enough to ravage an entire harvest leading to food insecurity (Hatfield and Prueger, 2015).

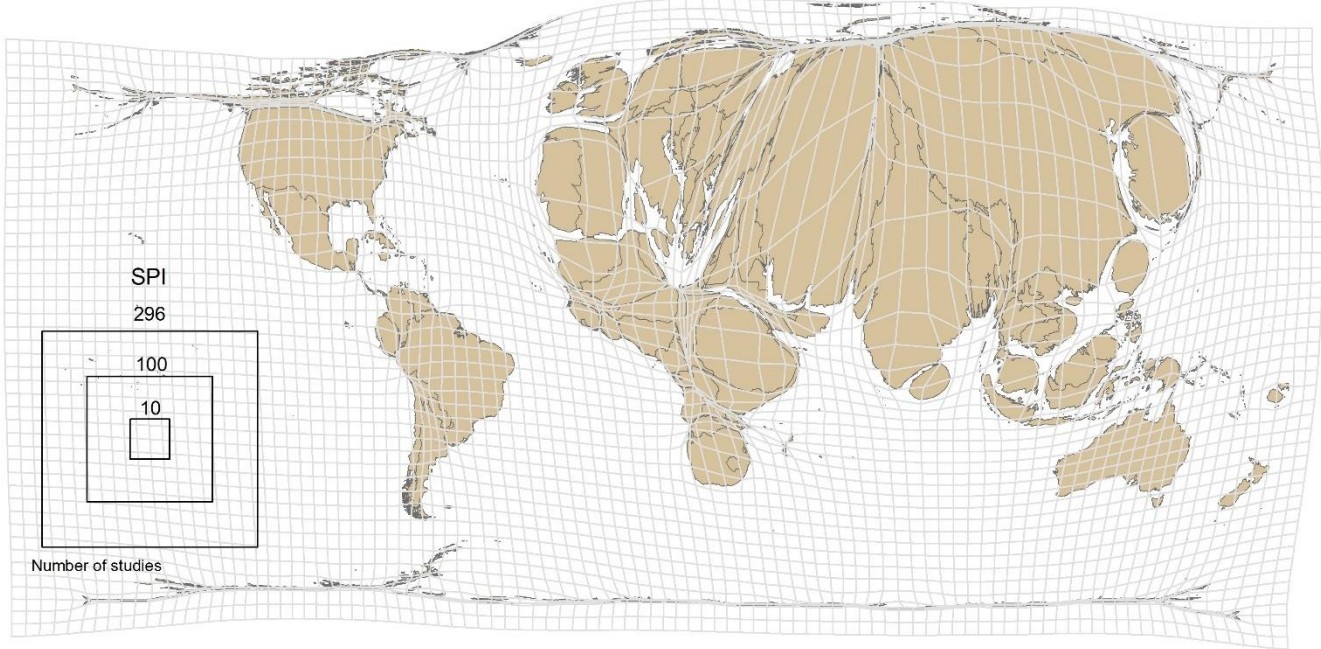

**Figure 6: Contiguous cartograms (Gastner-Newman) of the world with each country rescaled in proportion to the number of studies on Scopus related to drought and the SPI**

Most of the meteorological drought indices, beyond the SPI, are sensitive to the quantity and reliability of the data to fit the distribution. Their calibration requires a recommended 30 to 50 years of data. However, only very few regions of the world possess such an abundant historical hydrometeorological database. This is particularly challenging for developing countries.

According to the World Bank (2018), two thirds of the hydrological observation networks in developing countries are reported to be in poor or declining condition. The distribution of rain gauges across SSA is eight times lower than the WMO minimum recommended level, and while coastal West and Southern Africa, and the East Africa Highlands of Kenya and Uganda are

relatively well represented, areas of greater aridity are severely underrepresented (Walker et al., 2016). Consequently, reanalysis rainfall products are also less reliable for these more arid regions due to a lack of ground truthing data (Walker et al., 2016). The availability of data seems to be closely tied with the socio-economic condition of a country. As mentioned in Section 4.2, countries exposed to economic water scarcity generally experience a lack of capital to satisfy the demand for water and a lack of an extensive and well-maintained hydro-climatic monitoring network. Therefore, most of the countries of SSA are underrepresented or absent from publications related to drought indices, while high-income countries commonly report them (Fig.5).

The same applies for hydrological drought indices studies that are under-reported in SSA (Fig 3, 4 & 5). River flow monitoring networks in SSA are experiencing a similar decline to meteorological monitoring networks (Walker et al., 2016). However, globally, little attention seems to be given to the monitoring of hydrological drought indices (Fig. 1 &2). Long-term and regular hydrological monitoring is dependent of equipment and installations, their management and maintenance and the engagement of technical personnel. Not only hydrological monitoring is local and conditional by directly being related to the water supplies, but it requires high costs of implementation not always accessible for low and middle-income countries. In Europe, the lack of hydrological indices has been attributed more to a lack of a wide access and exchange of hydrometric data at regional, national and international scales due to economic, legal and practical barriers rather than a complete lack of related observations (Bachmair et al., 2016;Viglione et al., 2010).

In the Global North, data sharing is incentivised by funding bodies as an ongoing task alongside research activities. However, as Bezuidenhout and Chakauya (2018) highlight, funders operating in low and middle-income countries are not fully exploiting this power yet. But the main limitation goes beyond looser requirements or a lack of incentive by funders operating in low and middle-income countries concerning data sharing. In most African universities, promotion criteria are closely linked to publications of peer-reviewed journal articles (Bezuidenhout et al., 2017). Bezuidenhout and Chakauya (2018) stated that, the main, if not only, incentive, of researchers of many African universities to disseminate data is to publish it in peer-reviewed journals, which slows down its release rate. In the African continent, these limitations are compounded by questions of network density, data accessibility, temporal continuity, spatial representativeness, and tedious bureaucratic processes. These reasons led researchers investigating water resources dynamics in Africa to rely increasingly on modelled and satellite data (Hasan et al., 2019).

As Table 1 shows, NDVI – a remotely sensed index – is the most commonly used in agricultural drought-related studies. Only 3 out of the 15 agricultural drought indices are not remotely sensed. Just like the hydrological drought indices, this can reflect (i) the lack of hydrometric (field) observations or (ii) if they exist, a lack of sharing and access to them (Bachmair et al., 2016). Bachmair et al. (2016) highlight how "*the scarcity of water status observations, especially for groundwater, reflects the common focus on drought seen through the lens of rainfall and soil moisture that can be easily (remotely) monitored and/or modelled*". Indeed, the data needed to calculate agricultural drought indices seem more accessible. The most used index is the NDVI and requires land surface imagery containing both red and infrared bands and processing software; global NDVI datasets

are available open source at relatively high spatiotemporal distributions. As there are no requirements for historical data for calibration or a monitoring network, this could explain why the African continent more prominently reports agricultural drought than meteorological drought and hydrological drought (Fig. 5).

It is important to realise that data availability may be closely tied to the year of implementation of the drought indices. Indeed, hydro-climatic databases have different ages and dataset quality according to the country, but it can also be possible that the implementation of drought indices is a precursor of hydro-climatic data monitoring.

## 4.4 Scientific interest and orientation

As mentioned previously, in DEWS, the indices linked to the three categories of drought are seen as drivers as they are used to determine the occurrence and severity of a drought. However, as shown in Sections 3 and 4.3, the distinction between drought drivers and impacts, based on hydro-climatic variables, is context-dependent. First, the linear representation of drought implies that agricultural drought and hydrological drought are an impact of meteorological drought. Yet the indices used for meteorological drought have a different scope to those used for agricultural drought and hydrological drought. Taking the example of the most used indices, the SPI has a temporal focus with a strong statistical perspective on drought. Whereas for agricultural drought, the NDVI has a "spatial distribution" focus as it uses remote sensing to indirectly determine water-limitation in the vegetation at a specific time, like a snapshot of the vegetation health. In that sense, the NDVI measures a drought impact.

Moreover, water security is often confounded with hydrological drought. However, as we can see from Fig. 5d and Fig. 5e, the areas where each hydrological drought and water security are reported in scientific studies are not the same, suggesting that the occurrence of the first does not imply the other. In that sense, the literature seemingly indicates that hydrological drought is not the only driver of water security. It is well-established that human-driven demand affects water security, along with the hydrologic system (Van Loon et al., 2016a;Van Loon et al., 2016b).

The scientific reporting about drought suggests its risk of occurrence in an area and potentially an initiative of preparation for related damages. Though for each country, it is likely that drought is investigated according to: (i) a determined scientific approach, more physical or social; (ii) a purpose, in the sense of what is at greatest risk of being impacted by drought.

As shown in Table 1, most of the drought-drivers indices are investigated under the domain of environmental, Earth and agricultural sciences, suggesting a more physically-based approach. Food and water securities related to drought, respectively more reported in SSA and in Australia-Oceania (Fig. 5), are also studied through the scope of physical sciences but unlike the drivers, also through the lens of social sciences (Table 1).

Institutional incentives in many western countries may favour research that falls into well-defined silos. Research that meaningfully incorporates both physical and social science may not be sufficiently interesting to merit ground-breaking publications on both fronts; it may instead require one or the other discipline serving in a more consultative role.

Food security is a complex concept that requires a holistic approach. Food systems underpin food security and they are the result of the production, processing, distribution, preparation and consumption of food. These steps are themselves the results of dynamic interactions between and within the bio-geophysical and human environments (Gregory et al., 2005). Thus, its study requires the intervention of different specialists. Food systems encompass three main components : *"(i) food availability (with elements related to production, distribution and exchange); (ii) food access (with elements related to affordability, allocation and preference) and (iii) food utilisation (with elements related to nutritional value, social value and food safety)"* (Gregory et al., 2005). Hence, when food systems are stressed, food security is affected. As food security depends on many components, it stands vulnerable to the disturbance of any of them. These components can be disturbed by a range of factors that can be environmental, like droughts, but also circumstantial like conflict, changes in international trade agreements and policies, HIV/AIDS (Gregory et al., 2005). Food insecurity can be enhanced when these factors are combined. SSA is an area particularly prone to extreme heat-related impacts, as we mentioned in Sect. 4.2, but also to these circumstances. SSA holds : (i) more than 95% of farmed land relying on rainfed agriculture (Wani et al., 2009) ; (ii) about 75% of the world HIV/AIDS prevalence as of 2016 (Odugbesan and Rjoub, 2019);  (iii) 19 of the 43 economies with the highest poverty rate, all classified as in fragile and conflict-affected situations (Corral et al., 2020). This indicates that in drought-related studies focused on Sub-Saharan Africa, food security and the occurrence of these social processes is closely related.

Australia, known to be the driest inhabited continent (Hill, 2004), has a "*National Plan for Water Security*"(Government of Australia, 2007) that comprises a variety of mechanisms addressed by national and state governments (Cook and Bakker, 2012). Water security is also aimed to be addressed in an integrative and multi-scale way by "*taking action on climate change, using water wisely, securing water supplies and supporting healthy rivers and wetlands*"(Government of Australia, 2007).

Besides Australia, the fact that water security is reported for countries with extreme differences in socio-economics, such as countries in the Sahel and the USA (Fig. 5), suggests the experience of different types of water security. The definition of "water security" by UN Water (2013) is quite holistic. A population's access to adequate quantities of acceptable quality water has the goal to sustain three areas: livelihoods, human well-being, and socio-economic development (Montanari et al., 2013). Countries at different stages of development are more likely to focus on one of those three areas. Human well-being related to water-security can have many different understandings (Jepson et al., 2017;Hoekstra et al., 2018). Those can vary from one extreme to the other, as enough water for sanitary purposes, e.g. sanitation and showers, to indulgent leisure (E.g. swimming pools and gardens (Savelli et al., 2021;Bradley and Bartram, 2013;Willis et al., 2010)). In South Africa, experiences of Cape Town Day Zero's water crisis were diametrically different amongst the wealthy elite and the township dwellers. The first went through restrictions to water their garden and fill up their swimming pools while the second had insufficient water to take showers and go to the toilet (Savelli et al., 2021). Livelihoods and socio-economic development can also be understood and applied in different ways: from subsistence farming (Makurira et al., 2011) to agrobusiness and irrigation of crops meant for export (e.g. California (Morris and Bucini, 2016)).  The same can apply to food security: from malnutrition (Belesova et al., 2019) to the genetic adaptation of fruits and vegetable strains to droughts (Belesova et al., 2019;Basu et al., 2016).

Therefore, not only can areas be exposed to food and/or water insecurities, but they can be exposed to different declinations and severity within each. Water and food insecurities are very context specific, not even attributable to the country scale but to smaller areas. They are the result of complex and multi-disciplinary mechanisms, including social processes in addition to the physical ones. Thus, to be accurately monitored, drought-related water and food insecurities also need multi-disciplinary metrics. This comes in contradiction with drought indices that measure drought severity by looking only at the hydro-climatic

component. Consequently, by eluding (the monitoring of) social processes that can trigger and enhance drought impacts while solely focusing on their hydroclimatic component, DEWS seem to be formulating an incomplete forecast of the severity of droughts.

**4.5 Limitations**

The inability to deduce a cause-and-effect relationship between two variables, solely on the basis of an observed association

or correlation between them is common to all disciplines. The same applies for drought drivers and drought impacts even in drought prone areas. Drought and a related variable such as food security, may be directly related, or drought may be one of many stressors in a complex food system. Aligning a drought index and some type of impact variable is a good start but given the complexity of the systems in question, it is unlikely that drought would have sufficient explanatory or predictive power on its own. Without continuous and widespread monitoring of drought impacts, the societal pattern enabling understanding

of how drought is experienced differently and why, will not be identified. Therefore, the attempt of explaining the geographical repartition of drought-related impact studies by linking some features of drought to one or many of the four hypotheses detailed above, as per this study, remains then purely hypothetical.

Our approach separated studies by geography, principally at sub-continental scale. Other divisions on which to base our analysis could have been applied, like climatic or income levels, and may have led to additional insights. However, separating

studies by geographical region allowed highlighting of: (i) both physical and socio-economic similarities expected in homogenous; (ii) countries standing out. This enabled the investigation of potential justifications. Also, certain studies might be missing because they focus on regions rather than countries. We assume that this effect is fairly evenly distributed across the globe and consequently, we do not expect this to introduce a bias. Besides, for the majority of studies, the country (or countries) that (partly) coincides with the focus region is also mentioned in the title or abstract.

Disparities exist inside countries, particularly larger countries such as the United States, China, Brazil and India, where physical, socio-economic, data availability and interest disparities occur. However, because our drought indices and impacts investigation and analysis are at the country level, our discussion is also generalised to that scale. Getting rid of that aggregative propensity and grasping those regional disparities would have required an investigation at the scale of within-country regions (e.g.: California Central Valley, Brazilian semi-arid, the city of Cape Town). Yet, it is mostly the name of the countries that

are used in publications on Scopus. Moreover, that level of detail and analysis would be more appropriate for comparative studies between chosen semi-arid regions of the world rather than a broader study, like this one, where similar focus on drought and drought impacts indices are examined.

This study focuses on two types of drought-related impacts: food and water insecurities. Clearly, impacts of droughts are not limited to these two categories. For instance, text mining approaches conducted in Europe, based on media reports, showed that droughts lead to impacts related to forestry, fires, recreation, energy and transport sectors in addition to agriculture and water supply (Stahl et al., 2016;de Brito et al., 2020). The geographic distribution of the impact studies would be different if we also had considered impacts on, for instance, energy security, forestry, transport and tourism. Countries with predominant activities related to these sectors may have a high number of related drought impact studies, resulting in a different geographic repartition than the one shown in this present study. Our results are therefore only valid for the impact we evaluated: water and food securities.

The studies we obtained and analysed were a result of using Scopus, rather than another abstract and citation database, and of how we formulated our queries. Our search was constrained to articles having their title, abstract and keywords in English, potentially excluding important articles written in other languages. Additionally, the queries of the drought drivers were per indices, individually, while the queries of the impacts were regrouped by two themes. We justified the approach of grouping drought impacts keywords due to the lack of metrics existing for water and food insecurities related to drought, as it is the case for drought indices.

Also, working with word frequencies, as we did, could have led to the consideration of a drought index or impact that was only mentioned in the abstract as an example but that was not an object of the study. To verify this, we manually evaluated a random sample of 50 studies retrieved from Scopus. We did not identify any study mentioning a drought index while not using or investigating it. Concerning the impacts, we indeed found that sometimes, terms like "water security" (or other impacts or the key-words used in the related query detailed in the table A1) were utilised without being investigated in the study. However, for the cases that we encountered in our sample, the studies were global and had a more bibliographical scope. This means that no country was mentioned in the title, abstract or keywords. As mentioned in our methods section, we only considered studies mentioning a country in their title, abstract and keywords. This means that there is only a small chance that studies mentioning an impact without further investigating it were included in our analysis. They were generally discarded at an earlier stage because they do not mention any country.

Finally, we chose in our study to focus on how drought drivers and impacts were reflected in the scientific literature. However, disparities between topics of academic research and policy initiatives may exist. In addition, academic research may or may not align with other operational and ground truthed initiatives, such as efforts conducted by agencies and organisations working toward drought impacts relief, sustainable development and human welfare.

### 4.6 Recommendations

It has to be recognised and highlighted that DEWS have achieved the goal of providing timely and reliable information to decision makers for drought management and mitigation. As we aimed in our study to put drought-related variables in the appropriate context and appropriate relation to one another, we also acknowledge that the indices that DEWS rely on are

mostly conceptual and descriptive which contradicts DEWS operational purposes. The value of this study is to increase the relevance and utility of DEWS, which leads us to posit that their structure  tends to exclude the human influence on drought and drought influence on humans. The emphasis is on the natural effects on the hydrological system. Subsequently, the accuracy and efficiency of drought mitigation measures can be sub-optimal, based only on information lacking consideration of observed (local) drought impacts.

Several studies have promoted a shift of paradigm, aiming to define drought by its impacts and considering that if a system is impacted by a drought, this means that it was already vulnerable to drought (Blauhut et al., 2015;Blauhut et al., 2016). Analysing observed and inventoried past drought impacts across European countries was used as proxy to determine specific vulnerabilities. Dealing with drought may benefit from a diagnostic process that starts from analysing drought impacts rather than merely focusing on drivers (Walker, in press).

We recommend to also consider the human welfare aspects (e.g. food and water securities) that drought is affecting, rather than focusing on deficits of water volumes and flows only. In humanitarian approaches, a human welfare approach makes sense as the damages caused by a hazard and that aim to be addressed, can adversely affect, in the short and long-term, basic human safety through malnutrition, displacement, livestock or even human mortality. This approach is also applicable in drought management. Indeed, there is a lack of consensus in defining a drought and its impacts, resulting in difficulty in
agreeing on coherent and accurate drought metrics. Therefore, shifting the focus of drought mitigation to observable, graspable and quantifiable goals, such as human welfare, could overcome the uncertainty around drought and drought impacts definitions.

The human welfare proxy could be considered as an optimal situation without water shortage, e.g. zero hunger, poverty, conflicts and water insecurity. Thus, it could be aligned with the Sustainable Development Goals (SDG) as they (i) represent
the development priorities of both low- or high- income countries; (ii) benefit from existing and improvable metrics. Also, similarly to drought indices, SDGs have a global nature inclined to overlook the local context. By taking into account local particularities, the SDGs could be reached at the local level even if it is through a drought mitigation scope. Instead of the linear and still conceptual driver-focused "meteorological-agricultural-hydrological" droughts, the disaster scope could shift to more societally relevant goals linked to "poverty, water security, and food security". Thus, operational approaches of drought
management would be the equivalent of determining the extent to which drought is hampering the achievement of one or many of these defined goals. Therefore, our study calls for additional research analysing the role of drought in research on the Sustainable Development Goals, and more precisely about whether or not the DEWS are incorporated into development efforts by researchers.

Some studies have already been arguing in favour of considering other approaches than the two main top-down and bottom-
up approaches for climate change adaptation strategies (Ludwig et al., 2014;Conway et al., 2019). Both approaches come with their strengths and weaknesses and conciliating them represents a challenge and many complexities, often unsuitable for

integrating into water management (Ludwig et al., 2014). The issues complicating the decision-making are well known: the top-down approach is too broad and presents too much uncertainty; the bottom-up approach focuses too much on socio-economic vulnerability and too little on developing (technical) solutions (Ludwig et al., 2014). Thus, a risk-oriented approach

that focuses more on "*systems of receptors rather than conventional sectors*"(Warren et al., 2018), where research identifies vulnerability to different extreme events rather than only analysing their probabilities of occurence (Bliss and Bowe, 2011), is an alternative.

## 5 Conclusions

We conducted a bibliometric analysis on 5000+ scientific studies in which drought was associated to an index and water and

food securities, with the aim of comparing how drought drivers (e.g. precipitation, temperature, evaporative demand) and drought impacts (food and water insecurities) were reflected in the literature. Our results revealed that drought is mainly depicted through a focus on precipitation-based and remotely sensed indices. It is the SPI, a single-variable index, that is the most broadly used in different climatic and geographic contexts, despite being the one including the least local contextual information. Drought is regularly approached merely as a rainfall statistical anomaly and equated to meteorological drought.

Drought drivers studies tend to focus on particular geographical regions, especially northern countries, whereas studies reporting impacts related to food and water securities are more commonly located in Sub-Saharan Africa and Australia-Oceania respectively. Moreover, the areas where drought drivers are reported in scientific studies are different from the drought impacts ones. There is also a difference in the geographic repartition of drought-related food security and water security scientific studies. This suggests that drought-impacts studies are certainly dependent on both the physical and human processes occurring

in the geographic area, i.e. the local context.

Because "local context" can have different meanings, we raised four hypotheses that can be attributed to local context and that can contribute to drought drivers resulting in drought impacts. First, the physical availability of water; drought drivers indices measure the water deficit in one or several of the components of the hydrological cycle, implying that the severity of drought is the same in arid or humid climates. Second, the socio-economic conditions in the countries, as the income per capita and the

demography that affect, respectively, the capital involved in research and the vulnerability to hazards. Third, the data availability, related to the second point concerning socio-economic conditions, affects the selection and accuracy of an index, especially if the chosen index is unsuitable for the particular climate. Fourth, the scientific approach and the interest in the country that determines from which physical and/or social sciences scope drought will be looked at and for what purpose. It seems that drought impacts are considered more through social sciences lenses than drought drivers. Drought drivers indices

seem to remain conceptual metrics depicting climate features and do not seem to be linked to human-centred solutions. Also, both water and food securities are scientific concerns mostly in arid and semi-arid regions, from high to low income and whether drought drivers are investigated or not. This suggests many variants of the same type of impact according to what or who is likely to be most impacted by drought in the area.

Thus, more research is needed where the scope of drought mitigation is widened to the vulnerability to drought events rather

than only their probability of occurrence. DEWS would then more accurately predict the severity of a drought by also including

drought indices that are people-centred. In this way, drought metrics would also better align with SDSs. These drought metrics

could become more useful in monitoring the negative role of drought in achieving human welfare, and with that, the SDGs.

**Appendices**

| "M/A/H drought" Indices mentioned in the study | Acronym | Query |
|---|---|---|
| Standardized Precipitation Index | SPI | TITLE-ABS-KEY ( ( "Drought" ) AND ( "SPI" OR "Standardized Precipitation Index" ) ) |
| Standardized Precipitation Evapotranspiration Index | SPEI | TITLE-ABS-KEY ( ( "Drought" ) AND ( "SPEI" OR "Standardized Evapotranspiration Precipitation Index" ) ) |
| Aridity Index | AI | TITLE-ABS-KEY ( ( "Drought" ) AND ( "Aridity Index" ) ) |
| Precipitation Deciles | Deciles | TITLE-ABS-KEY ( ( "Drought" ) AND ( "Precipitation Decile*" OR "Rain decile*" OR "rainfall decile*" ) ) |
| Keetch-Byram Drought Index | KBDI | TITLE-ABS-KEY ( ( "Drought" ) AND ( "Keetch-Byram Drought Index" OR "KBDI" ) ) |
| Palmer Drought Severity Index | PDSI | TITLE-ABS-KEY ( ( "Drought" ) AND ( "Palmer Drought Severity Index" OR "PDSI" ) ) |
| Percent of Normal Precipitation (Index) | PNPI | TITLE-ABS-KEY ( ( "Drought" ) AND ( "Percent of Normal Precipitation" OR "Percent of Normal Precipitation Index" OR "PNPI" ) ) |
| Rainfall Anomaly Index | RAI | TITLE-ABS-KEY ( ( "Drought" ) AND ( "Rainfall Anomaly Index" OR "Rainfall Anomaly" OR "RAI" ) ) |
| Self-Calibrated Palmer Drought Severity Index | scPDSI | TITLE-ABS-KEY ( ( "Drought" ) AND ( "Self-Calibrated Palmer Drought Severity Index" OR "sc-PDSI" ) ) |
| Crop Moisture Index | CMI | TITLE-ABS-KEY ( ( "Drought" )AND ( "Crop Moisture index" OR "CMI" )) |
| Evaporative Stress Index | ESI | TITLE-ABS-KEY ( ( "Drought" ) AND ( "Evaporative Stress Index" OR "ESI" )) |
| Evapotranspiration Deficit Index | ETDI | TITLE-ABS-KEY ( ( "Drought" ) AND ( "Evapotranspiration Deficit Index" OR "ETDI" )) |

| Enhanced Vegetation Index | EVI | TITLE-ABS-KEY ( ( *"Drought"* ) AND ( *"Enhanced Vegetation Index"* OR *"EVI"* )) |
|---|---|---|
| Normalized Difference Vegetation Index | NDVI | TITLE-ABS-KEY ( ( "Drought" ) AND ("Normalized Difference Vegetation Index" OR "NDVI" )) |
| Leaf Area Index | LAI | TITLE-ABS-KEY ( ( *"Drought"* ) AND ( *"Leaf Area Index"* OR *"LAI"* )) |
| Palmer Moisture Anomaly Index – known as the Palmer Z index | PZI | TITLE-ABS-KEY ( ( *"Drought"* ) AND ( *"Palmer Z Index"* OR *"Palmer Moisture Anomaly Index"* OR *"PZI"* ) ) |
| Soil Adjusted Vegetation Index | SAVI | TITLE-ABS-KEY ( ( *"Drought"* ) AND ( *"Soil Adjusted Vegetation Index"* OR *"SAVI"* ) ) |
| Soil Moisture Anomaly | SMA | TITLE-ABS-KEY (( *"Drought"* ) AND ( *"Soil Moisture Anomaly"* OR *"SMA"* )) |
| Soil Moisture Deficit Index | SMDI | TITLE-ABS-KEY ( ( *"Drought"* ) AND ( *"Soil Moisture Deficit Index"* OR *"SMDI"* ) ) |
| Soil Water Deficit Index | SWDI | TITLE-ABS-KEY ( ( *"Drought"* ) AND ( *"Soil Water Deficit Index"* OR *"SWDI"* ) ) |
| Soil Water Storage | SWS | TITLE-ABS-KEY ( ( *"Drought"* ) AND ( *"Soil Water Storage"* OR *"SWS"* ) ) |
| Vegetation Condition Index | VCI | TITLE-ABS-KEY ( ( *"Drought"* ) AND ( *"Vegetation Condition Index"* OR *"VCI"* ) ) |
| Vegetation Drought Response Index | VegDRI | TITLE-ABS-KEY ( ( *"Drought"* ) AND ( *"Vegetation Drought Response Index"* OR *"VegDRI"* OR *"Veg DRI"* ) ) |
| Vegetation Health Index | VHI | TITLE-ABS-KEY ( ( *"Drought"* ) AND ( *"Vegetation Health Index"* OR *"VHI"* ) ) |
| Reservoir Level | | TITLE-ABS-KEY ( ( *"Drought"* ) AND ( *"Reservoir level\*"* OR *"water level in reservoir"* OR *"water levels in reservoirs"* ) ) |
| Palmer Hydrological Drought Index (PHDI) | PHDI | TITLE-ABS-KEY ( ( *"Drought"* ) AND ( *"Palmer Hydrological Drought Index"* OR *"PHDI"* ) ) |
| Streamflow Drought Index | SDI | TITLE-ABS-KEY ( ( *"Drought"* ) AND ( *"Streamflow Drought Index"* OR *"SDI"* ) ) |
| Standardized Runoff Index | SRI | TITLE-ABS-KEY ( ( *"Drought"* ) AND ( *"Standardized Runoff Index"* ) ) |
| Standardized Streamflow Index | SSFI | TITLE-ABS-KEY ( ( *"Drought"* ) AND ( *"Standardized Streamflow Index"* OR *"SSFI"* ) ) |
| Streamflow anomaly | | TITLE-ABS-KEY ( ( *"Drought"* ) AND ( *"streamflow anomaly"* ) ) |
| Standardized Water-level Index | SWI | TITLE-ABS-KEY ( ( *"Drought"* ) AND ( *"Standardized Water Level Index"* OR *"SWLI"* ) ) |

| Surface Water Supply Index | SWSI | TITLE-ABS-KEY ( ( *"Drought"* ) AND ( *"Surface Water Supply Index"* OR *"SWSI"* ) ) |
|---|---|---|
| **Drought impacts studies** | | |
| **Food security** | | TITLE-ABS-KEY("drought" AND ("food secur*" OR "food insecur*" OR "famine" OR "hunger" OR "hidden hunger" OR "malnourish*" OR "undernourish*" OR "malnutrition" OR "undernutrition" OR "crop loss*" OR "yield loss*" OR "agricultural loss*" OR "agricultural product* loss*" OR "loss of agricultural land*" )) |
| **Water security** | | TITLE-ABS-KEY ( ( *"drought"* ) AND ( ( ( *"safe"* ) AND ( *"water access"* OR *"drinking water"* ) ) OR ( ( *"clean"* ) AND ( *"drinking water"* OR *"drinking source"* ) ) OR *"freshwater availability"* OR *"water secur*"* OR *"water insecur*"* OR *"water crisis"* ) ) |

**TABLE A 1: Table of queries used in the advanced search of Scopus to retrieve the scientific studies of the drought indices and impacts.**

**Code and data availability**

Both code and data are available in the 4tu.ResearchData platform.
The doi and link of access is https://doi.org/10.4121/14452845.v2

**Author contribution**

SK has designed and conducted the research in collaboration with DWW, supervised by LAM and PRvO. SK has written the manuscript with input from all co-authors. The final version has been approved by all co-authors.

**Competing interests.**

Authors have declared no competing interests.

**Acknowledgements**

This work is part of the research program Joint SDG Research Initiative with project number 07.30318.016, which is (partly) financed by the Dutch Research Council (NWO) and the Interdisciplinary Research and Education Fund (INREF) of Wageningen University, the Netherlands.
We thank Germano Gondim Ribeiro Neto, Louise Cavalcante de Souza Cabral, Petra Hellegers and Eduardo Sávio Passos Rodrigues Martins that reviewed and provided helpful comments on earlier drafts of the manuscript.

We thank the four anonymous reviewers whose comments and suggestions helped improve and clarify this manuscript.

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
