# Peer review of "A geography of drought indices: mismatch between indicators of drought and its impacts on water- and food securities"

_Natural Hazards and Earth System Sciences, 2021_

## Author Response (AR1)

Reviewer 1

In this well-written manuscript the authors conduct a bibliometric analysis of papers that mention drought indices and water and food security concerns. The topic is interesting and meaningful. Furthermore, the methodology applied is robust, and the final outputs of good quality. Nevertheless, the title, abstract and introduction promise something bigger as they only mention "drought impacts". Forestry, energy, livestock and other impact types are ignored. The focus of the study should be clearer in the title and abstract. Another problem is that the results were not validated. There are many limitations that derive from using such a word frequency approach and they should be acknowledged in the limitations section. Furthermore, the discussion section should be deepened. Therefore, I suggest a major revision.

Dear reviewer,

Thank you very much for your positive and constructive review.

We agree that the initial title, the abstract and introduction mentioning "drought impacts" suggested an investigation of all potential impacts including those mentioned in the review that we didn't consider. Indeed, we focused only on food and water securities.

We therefore modified our title, abstract and introduction. We reformulated our title to "A geography of drought indices: mismatch between indicators of drought and its impacts on water- and food securities". We are now clearer that the focus is on food- and water-security-related impacts early in the abstract and introduction.

On behalf of all co-authors,

Sarra Kchouk

**Major comments**

- The authors should emphasize that only impacts related to water security and food security are considered. This should be very explicit in the title, abstract, introduction and discussion. This limits considerably the scope of the paper as impacts related to forestry, energy security, transport and even tourism are disregarded. Furthermore, this results in a bias in the outcome regions with a higher number of studies. E.g. if forestry was considered, I think there would be more studies in EUA and Europe

We agree and now clearly state that the focus is on food- and water securities-related impacts early in the abstract and introduction. In the discussion (L422), we added the following paragraph about the additional geographic bias introduced by only focusing on food- and water securities-related impacts.

"This study focuses on two types of drought-related impacts: food and water insecurities. Clearly, impacts of droughts are not limited to these two categories. For instance, text mining approaches

conducted in Europe, based on media reports, showed that droughts lead to impacts related to forestry, fires, recreation, energy and transport sectors in addition to agriculture and water supply (Stahl et al., 2016; de Brito et al., 2020). The geographic distribution of the impact studies would be different if we also had considered impacts on, for instance, energy security, forestry, transport and tourism. Countries with predominant activities related to these sectors may have a high number of related drought impact studies, resulting in a different geographic repartition than the one shown in this present study. Our results are therefore only valid for the impact we evaluated: water and food securities."

- Line 99: It should be further stressed in the introduction that a country comparison was conducted. I think the way the search was done can introduce biases. It could be that a studied investigated, West African countries for instance or that it investigated the Tyrol region or it could also be that just the name of a City/state were mentioned and not the country itself.

We agree. We included earlier in the introduction (L56) that the search was conducted based on the countries mentioned and not regions, cities or any other reference to geographic location.

"The aim of this study is to review scientific reporting on drought drivers and drought impacts for affected countries and analyse how these two compare. [...] We retrieved scientific studies from countries in which selected drivers of drought and food and water securities impacts of drought are mentioned. The components of drought drivers and impacts on which the literature focused were explored and compared for different areas of the world."

We also added his point in the discussion:

L411 : Also, certain studies might be missing because they focus on regions rather than countries. We assume that this effect is fairly evenly distributed across the globe and consequently, we do not expect this to introduce a bias. Besides, for the majority of studies, the country (or countries) that (partly) coincides with the focus region is also mentioned in the title or abstract.

And L417: Getting rid of that aggregative propensity and grasping those regional disparities would have required an investigation at the scale of within-country regions (e.g.: California Central Valley, Brazilian semi-arid, the city of Cape Town). Yet, it is mostly the name of the countries that are used in publications on Scopus.

- I would suggest making a correlation between the number of driver studies and the number of impact studies. You could either have a plot or a table with this for different countries or for different regions. This way we can visualize better the existing ratio. This will help to support your discussion. This could be used instead of Figure 4

We followed the reviewer's recommendations and made the suggested figures. They are below and represent: 1) a comparison between the number of drought indices studies and the number of drought impacts studies per country ; 2) the same but per region; 3) the value of the ratio of the number of studies of drought indices by the number of studies of drought impacts by region.

After careful consideration, we still believe that our current Figure 4 best conveys the message of our study. First, in the type of representation suggested by the reviewer, we aggregate the three

categories of drought indices as one (as drivers) and the two types of impacts as one, which overgeneralises the discussion. Our aim was to explain why there will be more focus in some countries on some type of drought indices and some other type of impacts, and we believe that this analysis is difficult to convey through the suggested type of figure.

Also, we believe that visualizing ratios could lead to errors of analysis. The ratios depend on how many studies have been published about each drivers or impacts. And there could be many reasons to that: the amount of researchers focusing on that topic; maybe in some countries, it is more common to publish many papers in the field of hydrology/meteorology than in the field of water and food securities; the unbalance due to the fact that we selected 34 drought indices and two categories of impacts; and the other reasons presented in our discussion section. We believe that the methodology we adopted was efficient to get rid of these bias to enable us to make a comparison between regions and between types of drivers and impacts. We standardised the number of studies by regions, thus correcting for the number of studies per regions. This led to the relative ratio of studies, which is better to compare to other regions.

For this reason, we decided to keep our current Figure 4 in our revised manuscript.

[Figure]

**Figure 1: Number of drivers and impacts studies per countries**

[Figure]

**Figure 2: Number of drivers and impacts studies per region**

[Figure]

**Figure 3: Ratio (drivers/impacts studies) per regions of the world**

- Section 4.1, besides the descriptive analysis, I think a more critical judgment is needed here. What are important indices that are hardly applied? Why they are hardly applied?

We thank the reviewer for the suggestions. We are explaining in this section (4.1), the preferred application of indices over others, justified by the physical conditions of the area. In the three following sections, we bring in other and deeper elements of analysis: the socio-economic conditions, the data availability and the scientific interest and orientation. We believe that these elements of analysis fall under what you suggest to further discuss. We also think that it is out of the scope of our study to analyse what important indices are hardly applied and why. This would imply investigating and discussing what indices can be considered as important, which is an interesting but different research question.

In our recommendation section (4.5), we suggest to use indices based on the SDGs. We hypothesize that these are hardly ever applied in drought research because there are multiple factors that influence these indices, drought being just one of them. This calls for a change of perspective compared to the current paradigm in the drought/DEWS community – to be impact-centred rather than drought-centred. As reviewer 3 formulated it: "what is the contribution of drought to food and water security? – rather than an assumed linearity between drought indices and food and water security."

- The discussion in 4.3 is directly related to 4.1 I would suggest to join them.

We agree that both sections are strongly related, with the second depending on the first, but not exclusively. We demonstrate that data availability is also a result of socio-economic conditions enabling countries, or not, to implement robust hydroclimatic monitoring networks. Also, both represent different problems that we believe are better discussed independently. Section 4.1 focuses on the climate and water resources of the area, whereas Section 4.3 (data availability) focuses on the capacity of a country for data collection and analysis. We decided to keep the order as it is as we believe that both Sections 4.1 and 4.2 lead to 4.3.

- A main limitation is that the definition of drought impact is very narrow. Important studies related to forestry in Europe and USA were completely ignored, for instance. In the recommendation section, if would be beneficial to add that future studies could look into impacts for other sectors. Examples of papers that give a good overview of potential impact types that could be investigated include:

https://iopscience.iop.org/article/10.1088/1748-9326/aba4ca

https://nhess.copernicus.org/articles/16/801/2016/

Thank you very much for this suggestion and for the references that we now utilise. The limitations section has been extended incorporating this information coupled with the first major comment of the review (L422).

"This study focuses on two types of drought-related impacts: food and water insecurities. Clearly, impacts of droughts are not limited to these two categories. For instance, text mining approaches conducted in Europe, based on media reports, showed that droughts lead to impacts related to forestry, fires, recreation, energy and transport sectors in addition to agriculture and water supply

(Stahl et al., 2016;de Brito et al., 2020). The geographic distribution of the impact studies would be different if we also had considered impacts on, for instance, energy security, forestry, transport and tourism. Countries with predominant activities related to these sectors may have a high number of related drought impact studies, resulting in a different geographic repartition than the one shown in this present study. Our results are therefore only valid for the impact we evaluated: water and food securities."

- A further limitation is that you work only with word frequencies. It could be that many of the impacts or indices are just mentioned in the abstract, but that the study does not really investigate it. For example, it could be that the abstract says "research is needed to help us understand food security". You could perhaps manually validate part of the abstracts and see what is the percentage of papers that fall into this category. In any case, this limitation should be clearly stated.

We agree and have added this clarification to the extended limitations section. We also conducted a manual check as suggested to evaluate how commonly indices and impacts may have been mentioned but not investigated.

Regarding the point about indices: we did not identify any studies mentioning an index and not using or investigating it.

Regarding the point about impacts: We found that indeed, sometimes terms like "water security" (or other impacts or the key-words used in the related query detailed in the table A1) were utilised without being the subject investigated in the study. However, when this was the case, the studies were global and had a more bibliographical scope rather than a case study scope. In these cases, no country was mentioned in the title, abstract or keywords. As mentioned in the last paragraph of our method section, l99, we only considered studies mentioning a country in their title, abstract and keywords. This mean that such studies, mentioning an impact with no further investigation, were not included in our analysis since they were discarded at an earlier stage.

We included all these considerations in a new paragraph L.423

"Also, working with word frequencies, as we did, could have led to the consideration of a drought index or impact that was only mentioned in the abstract as an example but that was not an object of the study. To verify this, we manually evaluated a random sample of 50 studies retrieved from Scopus. We did not identify any study mentioning a drought index while not using or investigating it. Concerning the impacts, we indeed found that sometimes, terms like "water security" (or other impacts or the key-words used in the related query detailed in the table A1) were utilised without being investigated in the study. However, for the cases that we encountered in our sample, the studies were global and had a more bibliographical scope. This means that no country was mentioned in the title, abstract or keywords. As mentioned in our methods section, we only considered studies mentioning a country in their title, abstract and keywords. This means that there is only a small chance that studies mentioning an impact without further investigating it were included in our analysis. They were generally discarded at an earlier stage because they do not mention any country."

- Here you try to link drivers and impacts by using simple linear approach. There are some quite interesting studies that actually try to link these data. I suggest adding a paragraph to the discussion session regarding the linkage of the physical aspects to the socio-economic ones. Some suggestion of studies that could be used to write this paragraph include:

https://hess.copernicus.org/articles/20/2779/2016/

https://www.nature.com/articles/s41467-019-12840-z

https://iopscience.iop.org/article/10.1088/1748-9326/10/1/014008/meta

It was by no means our intention to give the impression that drivers and impacts are linked in a linear fashion. Rather, we discuss in four different sections the important role of the local context, which hampers any generally applicable linear link between the two. We also suggested, as also specified earlier in this review, to change the perspective: start from the impact rather than the drought driver, because drought is just one of many drivers that leads to the final impacts. The suggested literature are indeed studies that aim to directly link drivers and impacts and are therefore of relevance to our manuscript. We added the following paragraph L455:

"Several studies have promoted a shift of paradigm, aiming to define drought by its impacts and considering that if a system is impacted by a drought, this means that it was already vulnerable to drought (Blauhut et al., 2015;Blauhut et al., 2016). Analysing observed and inventoried past drought impacts across European countries was used as proxy to determine specific vulnerabilities."

**Minor comments**

- Line 68: What is meant by "categorized" geographic areas?

"Categorised" was used to mean "grouped" in the sense that we grouped the countries according to geographic regions. We however changed this term for "countries", as suggested in one of the major comments and amended the sentence as follows (L59)

"We retrieved scientific studies from countries in which selected drivers of drought and food and water securities impacts of drought are mentioned."

- Line 75-76: The sentence is not clear. Metrics for what? Do you mean for selecting which indices were going to be reviewed?

Thank you for this comment. We meant by "metrics" the equivalent of "unit" and we reformulated it more clearly in the manuscript (l76).

"We considered the number of studies about drought indices and drought impacts, respectively, and their geographical distribution as our units."

- Table 1: please mention that this top 3 areas are retrieved from scopus

We agree and have amended the table as suggested. " "**Top 3 subject area retrieved from Scopus**"

- The Soil Moisture Index (SMI) is missing from the agriculture indices, or is it related to other of the mentioned indices? Please check. Here some references:

https://iopscience.iop.org/article/10.1088/1748-9326/11/7/074002/meta

https://hess.copernicus.org/articles/18/2485/2014/

Yes indeed, the SMI is missing from our listed agricultural drought indices. This is primarily because, as mentioned in our methodological section, we based our non-exhaustive list on two main publications: the IDMP handbook of drought indicators and indices (Svoboda and Fuchs, 2016); and a scientific study (Bachmair et al., 2016) where the authors gathered the most used drought indices in EWS by drought managers. In both publications, the SMI was not mentioned. We included this clarification in the methodology section, L77, as follows.

"Our list of drought indices is based on two prominent studies in the field of drought indices: commonly used indices to depict operational types of droughts (Svoboda and Fuchs, 2016) and the indices commonly used by water managers (Bachmair et al. (2016). Our list will, however, inherently be incomplete because many other indicators exist beyond the ones mentioned in these two studies."

Bachmair, S., Stahl, K., Collins, K., Hannaford, J., Acreman, M., Svoboda, M., Knutson, C., Smith, K. H., Wall, N., and Fuchs, B.: Drought indicators revisited: the need for a wider consideration of environment and society, Wiley Interdisciplinary Reviews: Water, 3, 516-536, 2016.

Svoboda, M. D., and Fuchs, B. A.: Handbook of drought indicators and indices, World Meteorological Organization Geneva, Switzerland, 2016.

Line 81-95: This information should come before the table 1

We agree but we were following the guidelines of the journal to the authors. It is recommended that figures and tables as well as their captions must be inserted in the main text near the location of the first mention.

- Figure 1: It is not possible to read some of the classes in the figure.

We have changed the dimensions of the figure and added leader lines.

[Figure]

- Line 122: I do not think there is a significant difference. For me they follow all the same pattern with some minor differences. "exceptions are Australia-Oceania and Sub-Saharan Africa, where AD indices are most frequently reported". Can you add confidence intervals to the plot?

Thank you for the comment and suggestion. We don't see an opportunity to include a confidence interval because what is represented is not a sample but all the studies we found by using the queries. However, we included a label mentioning the share of each type of MD/AD/HD studies, in percentage, to make the difference clearer.

- I am not convinced of using the acronyms MD, AD, HD. The terms "meteorological drought", "agricultural drought", etc are not so long, and I think it would be better for the reader to use them instead of MD, AD and HD

We now include the full terms rather than the acronyms throughout the manuscript: meteorological drought, agricultural drought and hydrological drought.

- Figure 2: If you opt to use the acronyms, it would be useful to add (MD), (AD), (HD) to the Figure. The acronyms are new and if you repeat then in the figures it makes it easier to read the text without needing to come back every time at the first time they were mentioned.

We have replaced all occurrences of these acronyms by the full terms in the text: meteorological drought, agricultural drought and hydrological drought

- Line 140: Again, I am not convinced of the use of SSA and similar acronyms. I had to go back in the text multiple times.

We prefer to use the acronyms for regions as those acronyms are frequently used in the literature. We have clearly introduced in the first lines of result sections (L120) the acronyms for the Middle-East and North Africa (MENA) and Sub-Saharan Africa (SSA). We removed many other

acronyms, based on the previous two suggestions, that should improve the readability already substantially.

- Figure 4: it is a nice plot, but does not add any new information. I suggest adding it to the supplementary material. I think this figure could be substituted by one where you show the ratio of driver vs impact studies.

We have already addressed this suggestion mentioned above in the major comments.

- Figure 1: I like the innovative visualizations, but I think a traditional choropleth map would convey information in an easier way

We assume the reviewer refers to Figure 5. We opted for a cartogram for many reasons. Firstly, it is an obvious way to eliminate a "visual" bias where there would be more studies because a country is larger. To eliminate this bias, the number of studies should be standardised to either the total number of studies (of MD, AD and HD) or the size of the country; which in our opinion introduces more uncertainty. Secondly, this type of visualisation is print and colour-blind friendly. A recent publication (Crameri et al., 2020) highlighted how colour maps visually distort data through uneven colour gradient and can be unreadable to those with colour-vision deficiency. Our aim was for the analysis and understanding of our figure to be independent from the used colours.

Crameri, F., Shephard, G. E., and Heron, P. J.: The misuse of colour in science communication, Nature Communications, 11, 5444, 10.1038/s41467-020-19160-7, 2020.

- Line 360: I would change "countries" by "regions", as for most of the analyses you aggregated the data.

Our analysis is mainly presented at the continental level, but our data was analysed at the country level. Therefore, we would like to keep referring to countries.

- Line 413-414: I do not think the results showed that "Our results revealed that drought is mainly depicted through a conceptual lens". I would remove the conceptual lens part as you have just focused on a word frequency and have not analysed the papers in detail

We removed this statement

- Line 358-363: This is not a limitation. The first sentences could be moved to the discussion above. The last sentences are a lot of speculation that should either be removed or backed up by other research

We agree that explaining the scale used for the search and the analysis of study (L408-410) is not a limitation *per se*. However, we would like to keep it where it is as we see it as a justification of the choice of the country scale to be the scale for our investigation and analysis, which is a limitation mentioned just before.

Previous Lines 362-363 about some regions inside countries centralising the work have been removed.

**Reviewer 2**

The authors try to give an overview of existing drought indices with respect to different focus aspects from different disciplines. This might be important as a large variety of drought definitions exist. Nevertheless, I have major concerns with the presented studies. The authors present a more or less pure bibliographic review of which indices exist, how often they are used in which part of the world, and for which purposes, which in my opinion is rather irrelevant for the scientific community. I was highly expecting more investigation like, where are the pros and cons of each index related to different aims and regions, where can they be applied, where are the limitations, which aspects are not covered by the existing indices, how can the indices be interpreted, so to say, what conclusions can be drawn out of the indices with respect for example for mitigation or adaptation strategies, and such aspects. It is definitely not clear to me, how to benefit from the presented study.

We thank the reviewer for the careful reading and comments aiming to improve the quality of our manuscript.

We acknowledge that the title we chose for our paper may have been suggestive of a different content than what was actually reported. We agree than the initial title mentioning "a review of drought indices" could be misleading, hence the unfulfilled expectations raised by the reviewer.

The study expected by the reviewer, i.e. "the pros and cons of each index related to different aims and regions" was not the focus of our paper. Our bibliographic search showed that the pros and cons of each index related to different aims have already been extensively covered in the literature (e.g. IDMP, 2016; Bachmair et al., 2016; Bachmair et al., 2015; Zargar et al., 2011; Sivakumar et al., 2011; Yihdego et al., 2019). We also believe that it would be difficult to summarise an in-depth analysis of drought indices by region in a brief paper. Instead, we targeted our study to review some of the most used drought indices for meteorological, hydrological and agricultural drought. With regard to drought impacts we decided to focus on impacts related to food security and water security.

IDMP (2016). Handbook of drought indicators and indices, (M. Svoboda and B.A. Fuchs). Integrated Drought Management Programme (IDMP) Tools and Guidelines Series 2. World Meteorological Organization (WMO), Geneva, Switzerland and Global Water Partnership (GWP), Stockholm, Sweden.

Bachmair, S., et al. (2016). "Drought indicators revisited: the need for a wider consideration of environment and society." Wiley Interdisciplinary Reviews: Water 3(4): 516-536.

Bachmair, S., et al. (2015). "Exploring the link between drought indicators and impacts." Natural Hazards and Earth System Sciences 15(6): 1381-1397.

Zargar, A., et al. (2011). "A review of drought indices." Environmental Reviews 19: 333-349.

Sivakumar, M. V., et al. (2011). Agricultural Drought Indices. Proceedings of an Expert Meeting: 2-4 June, 2010, Murcia, Spain, WMO.

Yihdego, Y., et al. (2019). "Drought indices and indicators revisited." Arabian Journal of Geosciences 12(3): 69.

Since we took a human-centred perspective, we believe that our findings contribute to the field of socio-hydrology and to the science of drought indices showing how local and contextual circumstances channel the choice of which drought indices to use. This constitutes for us the relevance for the scientific community.

We also posit in our study that the integration of local context and impacts into drought indices would add benefit, leading to more accurate drought monitoring. We believe that this is a significant outcome of our study and a conclusion that is applicable for mitigation and adaptation strategies, as the Reviewer mentions.

Therefore, we have reformulated our title to "A geography of drought indices: mismatch between indicators of drought and its impacts on water- and food securities". We believe this new title more accurately reflects the content of our study.

We have responded below to the individual comments of the Reviewer.

- Did you use the NHESS templates for the manuscript and bibliography? especially the references look weird

We did use the NHESS template available from their website for the manuscript and we used the reference style "Copernicus publications" (in EndNote) that we also downloaded from NHESS website. We have manually assessed the references and checked individually for any errors.

- Line 40ff. You state, that the initial driver for agricultural and/or hydrological droughts always comes from meteorology. I do not agree. You can have normal precipitation conditions but excessive land use and water extraction that could lead to AD and HD.

Typically, in the literature, the propagation of drought is depicted as a linear process rooting from meteorological drought and leading to AD and HD. We agree with your comment and re-wrote it more clearly, as follows (L35):

"The numerical value of hydro-climatic variables is associated to three main types of drought: meteorological, agricultural (or soil moisture) and hydrological droughts. These variables are in fact drivers, which refer to the contributing or counteracting factors that affect the development of droughts (Seneviratne, 2012). Those drivers are used by many drought studies as the framework to represent drought propagation. In the literature, the temporal propagation of drought is often considered to be a sequence occurring in an almost linear order (Wilhite and Glantz, 1985; Zargar et al., 2011; Bachmair et al., 2016), and in which humans have no direct influence. This is a simplification of a complex process, where it is considered that an anomaly (e.g. lower precipitation, higher temperature than average) of the values of those drivers will lead to a cascade reaction influencing the magnitude of other physical variables and leading in turn to the subsequent type of drought. As such, hydrological drought is inaccurately simplified as a result from the persistence in duration of agricultural (soil moisture) drought, which itself is simplistically attributed to the persistence of meteorological drought."

- For Table 1 you made a selection. Is there a chance to estimate the dark numbers of indices, that exist and that are not listed in this Table? I also miss relative indices like the effective drought index by Byun and Wilhite (1999). Furthermore, the table caption should be at the top of the Table not at the bottom. At the end of the Table some lines are highlighted in yellow, is there a reason for that?

Indeed, we have not included all existing drought indices, which would be a nearly impossible task. Rather, we selected those indices that have been listed in two main publications summarizing the most utilised drought indices. These two publications are: the IDMP handbook of drought indicators and indices (Svoboda and Fuchs, 2016); and a scientific study (Bachmair et al., 2016) where the authors gathered the most used drought indices in EWS by drought managers. This is mentioned in L77:

"Our list of drought indices is based on two prominent studies in the field of drought indices: commonly used indices to depict operational types of droughts (Svoboda and Fuchs, 2016) and the indices commonly used by water managers (Bachmair et al. (2016). Our list will, however, inherently be incomplete because many other indicators exist beyond the ones mentioned in these two studies."

It was an error to place the caption below the table. We have moved it to the top of the table.

The lines highlighted in yellow were aimed to attract the attention of the reader and emphasise how some categories of drought indices and the impacts groups were studied through the social science scope; a crucial point mentioned in our discussion. However, we have removed the highlight.

Bachmair, S., Stahl, K., Collins, K., Hannaford, J., Acreman, M., Svoboda, M., Knutson, C., Smith, K. H., Wall, N., and Fuchs, B.: Drought indicators revisited: the need for a wider consideration of environment and society, Wiley Interdisciplinary Reviews: Water, 3, 516-536, 2016.

Svoboda, M. D., and Fuchs, B. A.: Handbook of drought indicators and indices, World Meteorological Organization Geneva, Switzerland, 2016.

Line 100ff. The reasons how you select the studies and indices is definitely not clear to me

-We selected all the studies where the drought indices of Table 1 were mentioned in a drought-related study. The reasons why we selected these indices are the reasons mentioned in the answer to the previous Reviewer's comment (#3) and clarified in L76 as follows:

"We considered the number of studies about drought indices and drought impacts, respectively, and their geographical distribution as our units. Our list of drought indices is based on two prominent studies in the field of drought indices: commonly used indices to depict operational types of droughts (Svoboda and Fuchs, 2016) and the indices commonly used by water managers (Bachmair et al. (2016). Our list will, however, inherently be incomplete because many other indicators exist beyond the ones mentioned in these two studies. This resulted in 32 indices that we

linked to three main drought types (Table 1): meteorological (9 indices), soil moisture/agricultural (15) and hydrological (8) drought."

-The reasons why we selected these studies was to retrieve their (primary) country of application as explained in the methods section in L65 as follows:

"We investigated which indices of drought drivers are most frequently used in scientific drought-related studies and to what drought type they were linked. For each of these scientific studies we also retrieved the country of focus. This allowed us to identify: the most frequently mentioned type of drought for different geographic regions, and the prevalent drought indices used in scientific studies."

-The exact queries showing *how* we proceeded are available in Table A1 of the appendices and this is explained in L84 as follows:

"We opted for Scopus to retrieve the scientific publications of interest as it is the database covering the largest range of both, peer-reviewed literature type (scientific journals, books and conference proceedings), and disciplinary fields (science, technology, medicine, social sciences, and arts and humanities ) (Scopus, 2021). We then searched in the Scopus database for queries strictly including "drought" AND "[the indicator]" in the title, abstract and authors' keywords of the studies. We repeated the queries for each indicator individually as we were interested in knowing country-based preferences"

- For the figures like Fig.5, there should be a short explanation in the text of what this type of displaying represents and how to interpret such figures. Also, what do the squares in the bottom left corner of each figure stand for?

We added a short explanation of how to read the cartograms in the text L160:

"In this cartogram representation, each country has been rescaled in proportion to the number of studies on Scopus related to drought indices or water and food security impacts."

in the legend of the figure:

*"Contiguous cartograms (Gastner-Newman) of the world with each country rescaled in proportion to the number of studies on Scopus related to drought and a) Meteorological drought indices b) Hydrological drought indices c) Agricultural and Soil Moisture drought indices d) Food security e) Water security". The size of the square relates to the size of the countries and indicates the number of studies.*

We added the legend "number of studies" below those squares.

[Figure]

- There are some error messages in the text for missing references. Lines 241 and 278

  The error messages have been corrected.

- Lines around 250, I was expecting the total study more in this way like it is done for the SPI

  We decided to go into such detail for the SPI as it is globally the most used drought index due to requiring the fewest variables. Providing this level of detail for all the other indices would not be feasible in anything but an extremely lengthy paper. Also, this was beyond the aim of this paper, i.e. to explore the impacts component and the geography of both drivers and impacts.

This is a strong article that takes an interesting approach to understanding how and why drought research is framed differently in different places. It appears that the research team was systematic and methodical in their search of drought-related research. While search terms can always be adjusted, my sense is that they came up with results that are a fairly accurate representation of drought discourse on different continents related to food and water security. They note the limits of focusing on drought and its drivers as a physical phenomenon, and recommend focusing instead on aspects of human well-being, such as food and water supply. They accurately observe that this overarching focus on reducing poverty and improving food and water security would help resolve the issue of defining drought impacts. This larger framework, accounting for underlying vulnerability along with changing conditions, including drought, would put drought research in a larger context by asking, what is the contribution of drought to food and water security and to rates of poverty?

My main criticism is that some definitions or arguments may be tautological. I elaborate below. Also, to the authors' credit, they have at this stage explored many explanations. There is much good material here; it just needs some tightening and focus for greater clarity.

The analysis and description of how drought research is framed, or what its main questions are, by continent, is informative and solid.

Dear reviewer,

Thank you very much for your positive, constructive, and highly encouraging review, that certainly improved the quality of our manuscript.

Individual responses to the points you raised can be found below.

Comments related to findings:

- Line 221: In this paragraph or the next, it would be relevant to mention governance and/or corruption, some of the factors other than physical and demographic conditions that are widely recognized as contributing to food insecurity.

Yes, thank you for the useful suggestion. We have added the following paragraph at the end of section 4.2, L252:

"It is also important to mention the link between food security and governance. Food security is dependent on a complex interplay of factors. Some are outside the direct control of governments, like hydrometeorological extremes. But institutions, rules and political processes do play an important role in reaching increased food security. According to the Food and Agriculture Organization (FAO, 2011), "food security is unlikely to develop where there is not an organized, politically active and mobilized constituency pushing the issue higher on the public and political agenda". Thus, good governance is crucial for reaching food security. Corruption is one of the pervasive aspects of bad governance. It can affect food security by creating inefficiencies in the use of natural resources and food distribution (Economist Intelligence Unit, 2015). Practices of

corruption are spread in low-, middle- and high-income countries to different degrees (Transparency International, 2021) and in different levels of the food production and distribution chain (Transparency Int'l, 2019). Low-income countries are indeed the ones struggling the most to tackle corruption (Transparency International, 2021) contributing to their already prominent exposure to food insecurity. The addition of corruption, an indication of misallocation of resources and incapacity to successfully implement change and development, increases the risk of stagnation of food availability and indicates those countries as less suitable prospects for successful intervention (Economist Intelligence Unit, 2015)."

- Paragraph starting on 231: You could also conclude that focusing on physical drivers of drought is a "luxury," more apt to be of interest in places where more basic needs such as food security have been met.

Thank you very much, this is a very good point, which we have added to Section 4.2, L264:

"In other words, focusing on physical drivers of drought is an advantage more apt to be of interest in areas where more basic and essential needs, such as food security, have been met."

- Line 275: The sentence starting with "One reason …" is probably off the mark, overly concerned with definition, implying that people are somehow dismissing studying hydrologic drought because it's an impact and less worthy of study. It seems more likely that hydrologic monitoring is very local and conditional, directly related to water supplies, and data probably isn't shared, subject to the socio-economic conditions described in the preceding paragraph. Bottom line, researchers in those countries lack incentives and/or data to do the work.

We agree with the reviewer and we amended the paragraph to include the suggestions accompanied by bibliographic references, as follows (L301):

"Long-term and regular hydrological monitoring is dependent of equipment and installations, their management and maintenance and the engagement of technical personnel. Not only is hydrological monitoring local and conditional by being directly related to the water supplies, but it requires high costs of implementation that are not always accessible for low and middle-income countries. In Europe, the lack of hydrological indices has been attributed more to a lack of wide access and exchange of hydrometric data at regional, national and international scales due to economic, legal and practical barriers rather than a complete lack of related observations (Bachmair et al., 2016; Viglione et al., 2010).

In the Global North, data sharing is incentivised by funding bodies as an ongoing task alongside research activities. However, as Bezuidenhout and Chakauya (2018) highlight, funders operating in low and middle-income countries are not fully exploiting this power yet. But the main limitation goes beyond looser requirements or a lack of incentive by funders operating in low and middle-income countries concerning data sharing. In most African universities, promotion criteria are closely linked to publications of peer-reviewed journal articles (Bezuidenhout et al., 2017). Bezuidenhout and Chakauya (2018) stated that, the main, if not only, incentive, of researchers of many African universities to disseminate data is to publish it in peer-reviewed journals, which slows down its release rate. In the African continent, these limitations are compounded by questions of network density, data accessibility, temporal continuity, spatial representativeness, and tedious bureaucratic

processes. These reasons led researchers investigating water resources dynamics in Africa to rely increasingly on modelled and satellite data (Hasan et al., 2019).

- Line 291: I usually think of a "driver" as a meteorologic or physical system or condition that creates drought, which is measured by a drought index.

We agree with the reviewer and believe a typo created this confusion. Earlier in the text (L36) we provide a definition matching the reviewer's suggestion. We reformulated the sentence to read: "The indices linked to the three categories of drought are seen as drivers as they refer to the contributing or counteracting meteorological or physical factors that affect the development of droughts. They are used to determine the occurrence and severity of a drought. However..."

- Line 293: Rather than saying it is not clear, perhaps state that it is context-dependent.

We agree with the reviewer and rephrased the sentence as suggested, L334: "However, as shown in Sections 3 and 4.3, the distinction between drought drivers and impacts, based on hydro-climatic variables, is context-dependent."

- Line 300: It is well-established that human-driven demand affects water security, along with the hydrologic system. You could say this is consistent with Van Loon et al 2016.

Thank you for the suggestion. We have included this reference to strengthen our statement as follows, L345:

It is well-established that human-driven demand affects water security, along with the hydrologic system (Van Loon et al., 2016a; Van Loon et al., 2016b).

- Line 320: The sentence starting on 321 is a bit of a contortion. How would food security NOT be related to these social processes? And what is "food security related to drought studies"? Food security as seen through the lens of drought studies? Just trying to construct this sentence suggests that too much focus on drought obscures the larger goal.

We tried to tie the discussion as much as possible to our methodology which was based on retrieving studies mentioning "drought" and the keyword of interest. Thus, by "food security related to drought studies", we meant the studies mentioning "drought" and an impact linked to food-security. We rephrased in L369 as:

"This indicates that in drought-related studies focused on Sub-Saharan Africa, food security and the occurrence of these social processes is closely related."

- Line 346: It's not drought indices that are eluding monitoring of social processes that contribute to impacts. It's the focus of inquiry or intent. A drought index is one thing. Variables or indices related to food or water security are another piece of it. There may be many pieces in a bigger system.

Thank you for the suggestion, we agree. The focus here was on drought indices while it should have been on the monitoring systems that only use this type of information. What contributes to an incomplete prediction of drought impacts and their severity is the omission in those monitoring systems of indicators depicting social processes. We rephrased it as follows L394:

"Consequently, by eluding (the monitoring of) social processes that can trigger and enhance drought impacts while solely focusing on their hydroclimatic component, DEWS seem to be formulating an incomplete forecast of the severity of droughts."

- Line 351: Drought and a related variable such as food security may be directly related, or drought may be one of many stressors in a complex food system. Aligning a drought index and some kind of impact variable is a good start but given the complexity of the systems in question it is unlikely that drought would have sufficient explanatory or predictive power on its own. I think this is actually what you are saying but the final sentence of this paragraph is a bit murky.

Yes, that is the point that we are attempting to make. We have rephrased this paragraph according to your suggestion that seems to convey the main message in a clearer way. L398:

"The inability to deduce a cause-and-effect relationship between two variables, solely on the basis of an observed association or correlation between them is common to all disciplines. The same applies for drought drivers and drought impacts even in drought prone areas. Drought and a related variable, such as food security, may be directly related, or drought may be one of many stressors in a complex food system. Aligning a drought index and some type of impact variable is a good start but given the complexity of the systems in question, it is unlikely that drought would have sufficient explanatory or predictive power on its own. Without continuous and widespread monitoring of drought impacts, the societal pattern enabling understanding of how drought is experienced differently and why, will not be identified."

- Line 360: Final sentence may not be needed.

Yes, thank you for the suggestion. We deleted it.

- Line 363: "might be centralizing the background work" ??? reword, please

Yes, this was speculation that Reviewer 1 also pointed out. We deleted this sentence.

- Line 364: This is an extreme understatement.

We agree and believe that it is the conditional form used in the first submission that led to the understatement. We amended the paragraph as follows, L414:

"Disparities exist inside countries, particularly larger countries such as the United States, China, Brazil and India, where physical, socio-economic, data availability and interest disparities occur."

- Lines starting with 378: Clarify this paragraph. Is this study about increasing the relevance and utility of drought-related variables? Or about framing questions that put drought-related variables in appropriate context, and appropriate relation to one another?

Indeed, there was a mistake in line 380 of the first submitted version of the manuscript. It is not about drought-related variables but drought monitoring systems mainly relying on physically-based indices. We have corrected this and incorporated your suggestion of mentioning that our study also contributes to "put drought-related variables in appropriate context, and appropriate relation to one another". We amended the paragraph as follows (L446):

"It has to be recognised and highlighted that DEWS have achieved the goal of providing timely and reliable information to decision makers for drought management and mitigation. As we aimed in our study to put drought-related variables in the appropriate context and appropriate relation to one another, we also acknowledge that the indices that DEWS rely on are mostly conceptual and descriptive which contradicts DEWS operational purposes. The value of this study is to increase the relevance and utility of DEWS, which leads us to posit that their structure tends to exclude the human influence on drought and drought influence on humans. The emphasis is on the natural effects on the hydrological system. Subsequently, the accuracy and efficiency of drought mitigation measures can be sub-optimal, based only on information lacking consideration of observed (local) drought impacts..."

- Paragraph starting on 392: Yes, yes, yes! … Does this suggest the basis for a next bibliometric study, analyzing the role of drought in research on Sustainable Development Goals? Do your results shed new light on how researchers are or are not incorporating DEWS into development efforts?

At this stage, we don't believe we can directly draw conclusions based on our results on how DEWS are part of development efforts. Therefore, we have amended this paragraph in order to provide suggestions for future research, as follows L475:

"Therefore, our study calls for additional research analysing the role of drought in research on the Sustainable Development Goals, and more precisely about whether or not the DEWS are incorporated into development efforts by researchers."

- Line 385: Or thinking bigger than drought mitigation, to mitigation of food and water insecurity, in which drought plays a role. You have this in the sentence starting on Line 390, but take a convoluted route to get there.

We agree and have amended the paragraph as follows (L459):

"We recommend to also consider the human welfare aspects (e.g. food and water securities) that drought is affecting, rather than focusing on deficits of water volumes and flows only. In humanitarian approaches, a human welfare approach makes sense as the damages caused by a hazard and that aim to be addressed, can adversely affect, in the short and long-term, basic human safety through malnutrition, displacement, livestock or even human mortality. This approach is also applicable in drought management. Indeed, there is a lack of consensus in defining a drought and its impacts, resulting in difficulty in agreeing on coherent and accurate drought metrics. Therefore, shifting the focus of drought mitigation to observable, graspable and quantifiable goals, such as human welfare, could overcome the uncertainty around drought and drought impacts definitions."

- Could mention somewhere: Institutional incentives in many western countries may favor research that falls into well-defined silos. Research that meaningfully incorporates both physical and social science may not be sufficiently interesting to merit ground-breaking publications on both fronts; it may instead require one or the other discipline serving in a more consultative role.

Thank you for this suggestion. We agree and included this exact paragraph in our section 4.4. Scientific Interest and Orientation, at line 353.

Minor:

- Line 182: Either delete "a" or the "s" on "events".

Thank you; we deleted the "a".

- Line 201: "dry" not "dryer"

Thank you; we corrected it.

- Line 230, probably "population" instead of "demography"

Yes, we corrected it to population (growth).

- Line 318: Extra word?

Yes, "heatwaves" has been deleted.

**Reviewer 4**

The paper addresses an important topic and provides new information based on an innovative approach to analyze the usage of drought indices. The authors used the Scopus database to review several aspects of the use of these indices in the scientific literature. Drought indicators are frequently used not only in a scientific context but also for several practical applications. The results are therefore worthy to be published, but I have one concern related to methodology and conclusions.

Reviewer 1, 2, and 3 already gave a number of detailed comments and I am therefore not going into such details again.

If I understand the methodology and database queries correctly, the authors ignored the temporal development in the use of the indices. In section 2.2 the authors explain that the articles were published between 1960 and 2021. I assume that within this period there have been significant changes in the usage of the indices. Such changes might have been caused by improved data availability (either due to changes in data policies, data exchange, or new observing systems (e.g. availability of satellite data)), but changes could also be related to scientific progress or a change in the societal view on droughts (i.e., increased security issues in some regions or decreased risks due to improvements in the related infrastructure, etc.). Population, income, and economic activities have significantly changed in many regions over that period. It might be possible that some indices have been used more often in the past, e.g., those that require only a small availability of input data.

In my understanding, the results and conclusions do therefore not necessarily represent the current practice in the use of the indices. Some of the recommendations of the authors might therefore already been implemented, but if that is the case, the reporting on that might just be part of the most recent literature and therefore only be mentioned in a small number of papers.

I would therefore suggest that the authors provide some additional remarks to what extend the results reflect the current practice and if there are indications for a change during the decades since 1960. If the authors have indications that there is no significant change over time then this should be explained in the discussion sections. Otherwise, restricting the database queries to selected decades could be an approach to illustrated changes in the usage statistics.

Dear reviewer,

Thank you very much for your positive and constructive review.

We will here respond to the main concern raised.

During the preliminary research that led to the results mentioned in our study, we did indeed conduct a time analysis. We visualised and compared the evolution of the usage of drought indices and drought impacts in the literature in order to analyse and link it to the same factors that you mentioned. However, we did not find any remarkable pattern, peak or correlation. Therefore, we decided to not include this part. The graph of the evolution is attached below.

[Figure]

We do recognise, however, that the matter of temporal evaluation should have been mentioned in our manuscript as it is a concern that may arise. We have incorporated your suggestion in our result section L133 by mentioning that a time-analysis of the usage of the indices has been done and no clear changes in reported indices over time were observed, as follows:

"During the preliminary research that lead to the results mentioned in our study, we conducted a time analysis. We visualised and compared the evolution of the usage of drought indices and drought impacts in the literature in order to analyse and link it to factors such as improved data availability, scientific progress or a change in the societal view on droughts (not shown). However, we did not find any remarkable pattern, peak or correlation. Therefore, we decided to not include this part in our study."

---

## Referee Report (RR1)

Line 57: For the U.S. example, you may want to refer to the U.S. Drought Monitor, a specific product, rather to than the National Drought Mitigation Center, which has developed many products, including several that focus on detecting impacts independently of drought status (see droughtimpacts.unl.edu). However, impacts is one of many inputs to the U.S. Drought Monitor. You may want to add a clarifying statement explicitly stating that underlying vulnerabilities are not taken into account in most DEWS.

Later in discussion 4.4 you may want to add that while most DEWS don't take vulnerability into account, in contrast FEWS (Famine Early Warning System) does consider vulnerabilities. A DEWS is broad-spectrum and weather-driven, warning of drought and letting various sectors respond as they are able. This seems consistent with its conceptual origins, coming from a meteorological services perspective. The concept of "drought impacts" may be too broad to be fully accounted for in a DEWS. It takes a more purpose-built system to express the relationships between an impact, underlying vulnerabilities and physical drought. FEWS originates from a humanitarian perspective and has a more specific purpose, preventing or mitigating famine.

Lines 59 & 60: Can you reword to avoid referring to "aims" in two consecutive sentences, and instead make it clearer that the ideas in the second sentence flow from the first?

Line 81: Reword to "indices commonly used operationally to depict different types of drought"

Line 167: How about "larger than" rather than "superior to", more in keeping with statistical terminology

Line 449 – Can probably delete either "drivers" or "indices."

Under "limitations," you may also want to mention the possible disparity between topics of academic research and policy initiatives. Academic research may or may not align with "real-world" initiatives, such as efforts by agencies and organizations that are working toward Sustainable Development Goals.

---

## Author Response (AR2)

The scope of this paper is ambitious and it provides many interesting insights into research on drought indicators and impacts, world-wide. The figures are interesting and informative. My revision suggestions at this stage are relatively minor, mainly having to do with 1) whether drought impacts figure into the depiction of drought in the U.S., and 2) the possible need to distinguish between what researchers study and what is actually being done or used by agencies and organizations working toward sustainable development goals.

Dear reviewer,

Thank you very much for your positive and constructive review. Individual responses to the points you raised can be found below.

- Line 57: For the U.S. example, you may want to refer to the U.S. Drought Monitor, a specific product, rather to than the National Drought Mitigation Center, which has developed many products, including several that focus on detecting impacts independently of drought status (see droughtimpacts.unl.edu). However, impacts is one of many inputs to the U.S. Drought Monitor. You may want to add a clarifying statement explicitly stating that underlying vulnerabilities are not taken into account in most DEWS.

Yes, thank you for this clarification. We have amended the paragraph as follows L53:

"In addition, most the of DEWS do not take the underlying vulnerabilities of the drought affected or monitored areas into account. Thus, in the current configuration of most DEWS, the presumed likelihood of experiencing impacts is mainly linked to the severity of climatic features only (e.g. Princeton Flood and Drought Monitors; U.S. Drought Monitor; Brazilian Drought Monitor)."

- Later in discussion 4.4 you may want to add that while most DEWS don't take vulnerability into account, in contrast FEWS (Famine Early Warning System) does consider vulnerabilities. A DEWS is broad-spectrum and weather-driven, warning of drought and letting various sectors respond as they are able. This seems consistent with its conceptual origins, coming from a meteorological services perspective. The concept of "drought impacts" may be too broad to be fully accounted for in a DEWS. It takes a more purpose-built system to express the relationships between an impact, underlying vulnerabilities and physical drought. FEWS originates from a humanitarian perspective and has a more specific purpose, preventing or mitigating famine.

We thank the reviewer for this suggestion. Indeed, FEWS take into account vulnerability factors as famine is closely tied to those. We believe it is relevant to highlight the importance of taking into account drought impacts and amongst them, food insecurity. The occurrence of famine results from a combination of many factors among which drought-related aspects play a relevant role indeed. Since our focus is on drought and DEWS specifically we decided to not take FEWS onboard in our analysis. Rather than principally weather-driven we prefer a drought paradigm that includes a broad range of drought impacts well beyond a narrow meteorological view.

We have however added the following clarification in our Recommendations section, L63

"Dealing with drought may benefit from a diagnostic process that starts from analysing drought impacts rather than merely focusing on drivers (Walker, in press)".

Walker, D., Cavalcante, L., Kchouk, S., Ribeiro Neto, G., Dewulf, A. Gondim, R., Martins, E., Melsen, L., Souza Filho, F., Vergopolan, N., Van Oel, P.: Drought diagnosis: what the medical sciences can teach us. In: Earth's Future, in press.

- Lines 59 & 60: Can you reword to avoid referring to "aims" in two consecutive sentences, and instead make it clearer that the ideas in the second sentence flow from the first?

Thank you for this suggestion. We have amended the paragraph as follows, L57:

"This study aims to review scientific reporting on drought drivers and drought impacts for affected countries and analyse how these two compare. Improving our understanding of the linkage and separation between drought drivers and drought impacts enables us to provide directions to further improve the accuracy of the information provided by DEWS."

- Line 81: Reword to "indices commonly used operationally to depict different types of drought"

Thank you for this suggestion. We have reworded the sentence as suggested L79

- Line 167: How about "larger than" rather than "superior to", more in keeping with statistical terminology

Yes, thank you for the suggestion. We replaced it as suggested L157.

- Line 449 – Can probably delete either "drivers" or "indices."

We believe that it was meant L436. Thank you for the suggestion. We deleted "drivers".

- Under "limitations," you may also want to mention the possible disparity between topics of academic research and policy initiatives. Academic research may or may not align with "real-world" initiatives, such as efforts by agencies and organizations that are working toward Sustainable Development Goals.

Thank you very much for your suggestion. The Sustainable Development Goals are introduced, as a recommendation for the orientation of drought indices, only in the next Recommendation section. Thus we decided to mention instead "sustainable development and human welfare". We amended the paragraph in the end of our Limitations section, as follow L447:

"Finally, we chose in our study to focus on how drought drivers and impacts were reflected in the scientific literature. However, disparities between topics of academic research and policy initiatives may exist. In addition, academic research may or may not align with other operational and ground truthed initiatives, such as efforts conducted by agencies and organisations working toward drought impacts relief, sustainable development and human welfare."